# Pyridoxine induces glutathione synthesis via PKM2-mediated Nrf2 transactivation and confers neuroprotection

Yao Wei[1,4], Ming Lu[2,4], Meng Mei[1,4], Haoran Wang[2], Zhitao Han[1], Miaomiao Chen[2], Hang Yao[2], Nanshan Song[2], Xiao Ding[1], Jianhua Ding[2], Ming Xiao[2] & Gang Hu[1,2,3 ✉]

Oxidative stress is a major pathogenic mechanism in Parkinson's disease (PD). As an important cellular antioxidant, glutathione (GSH) balances the production and incorporation of free radicals to protect neurons from oxidative damage. GSH level is decreased in the brains of PD patients. Hence, clarifying the molecular mechanism of GSH deficiency may help deepen our knowledge of PD pathogenesis. Here we report that the astrocytic dopamine D2 receptor (DRD2) regulates GSH synthesis via PKM2-mediated Nrf2 transactivation. In addition we find that pyridoxine can dimerize PKM2 to promote GSH biosynthesis. Further experiments show that pyridoxine supplementation increases the resistance of nigral dopaminergic neurons to 1-methyl-4-phenyl-1,2,3,6-tetrahydropyridine (MPTP)-induced neurotoxicity in wild-type mice as well as in astrocytic Drd2 conditional knockout mice. We conclude that dimerizing PKM2 may be a potential target for PD treatment.

[1] Department of Pharmacology, Nanjing University of Chinese Medicine, Nanjing, Jiangsu 210023, China. [2] Jiangsu Key Laboratory of Neurodegeneration, Nanjing Medical University, Nanjing, Jiangsu 210029, China. [3] Biomedical Functional Materials Collaborative Innovation Center, College of Chemistry and Materials Science, Nanjing Normal University, Nanjing, Jiangsu 210023, China. [4]These authors contributed equally: Yao Wei, Ming Lu, Meng Mei. ✉email: ghu@njucm.edu.cn

Parkinson's disease (PD), which is characterized by a profound loss of substantia nigra pars compacta (SNc) dopaminergic neurons, is the most common neurodegenerative movement disorder[1,2]. Oxidative stress, defined as the overproduction of reactive oxygen species (ROS), is widely considered a major pathogenic mechanism of PD for at least three reasons. First, substantial oxidative damage to nucleic acids, lipids, and proteins has been detected in many PD models as well as in the SNc of PD patients[3–5]. Second, not only environmental factors such as 1-methyl-4-phenyl-1,2,3,6-tetrahydropyridine (MPTP) and rotenone but also familial genes such as *SNCA* (PARK1), *Parkin* (PARK2) and *DJ-1* (PARK7) lead to PD pathogenesis partially by inducing overwhelming ROS production[6–8]. Third, dopaminergic neurons have increased vulnerability to oxidative stress because they produce a large amount of dopamine, which can be oxidized by ROS, subsequently resulting in lysosomal dysfunction and α-synuclein accumulation[9]. Hence, the detoxification of brain ROS is an essential task for PD treatment.

Glutathione (GSH), a ubiquitous thiol tripeptide, is an important antioxidant in the brain[10,11]. Glutamate-cysteine ligase (Gcl) is the rate-limiting enzyme in GSH synthesis and condensing glutamine and cysteine into γ-glutamylcysteine, which is then combined with glycine in a reaction catalyzed by GSH synthase to form GSH[10]. GSH serves as an electron donor to reduce ROS and is oxidized into glutathione disulfide (GSSG). GSSH can then be regenerated to GSH by acquiring electrons from NADPH through a reaction catalyzed by glutathione reductase (GR)[12]. Thus, the total GSH level in the brain determines the ROS buffering capacity and protects neurons from oxidative damage. Unfortunately, accumulating evidence indicates that GSH is greatly decreased in the brains of PD patients, increasing the vulnerability of neurons to oxidative stress[13–16]. Hence, clarifying the mechanism underlying the GSH decline and trying to restore the GSH level are very important in PD treatment.

As antiparkinsonian medications, dopamine agonists have been reported to induce GSH biosynthesis in the brain[17–19], indicating that dopamine receptors modulate GSH synthesis. Dopamine receptors are expressed in both neurons and glial cells[20–22]. However, functional research on dopamine receptors has focused on neurons because dopamine is a well-known neurotransmitter. The function of astrocytic dopamine receptors was long overlooked until we recently reported that activation of the astrocytic dopamine D2 receptor (DRD2) can suppress neuroinflammation in PD[23,24]. Because GSH is mainly synthesized in astrocytes in the brain[25–27] and dopamine receptors on astrocytes are functional, we hypothesized that astrocytic dopamine receptors regulate GSH synthesis.

Pyruvate kinase (PK) is the final rate-limiting enzyme in glycolysis and catalyzes the conversion of phosphoenolpyruvate to pyruvate[28]. The four PK isoforms in mammals are encoded by two genes. The *PKLR* gene encodes PKL and PKR, which are expressed in the liver and erythrocytes, respectively, and the *PKM* gene encodes PKM1 and PKM2, both of which are expressed in various types of cells and tissues[29]. In recent years, in addition to its pyruvate kinase function, PKM2 has sparked considerable interest for its noncanonical function, which regulates gene expression as a transcription factor coactivator of Hif-1α, β-catenin, and p53[30–32] or phosphorylates proteins as a protein kinase[33–36].

In the current study, with the help of pharmacological methods and animal models, we demonstrate that astrocytic DRD2 induces GSH synthesis via PKM2-mediated Nrf2 transactivation. Our results indicate a vicious cycle of decreased dopamine release—DRD2 signaling deficiency—decreased GSH synthesis—oxidative stress—dopaminergic neuron death in the progression of PD. We

selecte PKM2 as a target to restore GSH levels and further find that pyridoxine can dimerize PKM2 to promote GSH biosynthesis. Animal experiments show that pyridoxine treatment increases the resistance of nigral dopaminergic neurons to MPTP-induced neurotoxicity in wild-type mice as well as in astrocytic *Drd2* conditional knockout mice. These findings uncover a new function of PKM2 and provide a potential therapeutic strategy for PD treatment beyond targeting dopamine receptors.

## Results

**Astrocytic DRD2 facilitates GSH synthesis via Nrf2 activation.** To study whether dopamine receptors on neurons or astrocytes induce GSH synthesis, we treated primary neurons and astrocytes for 24 h with the typical dopamine agonist cabergoline, which has been reported to increase brain GSH levels[19]. Different concentrations of cabergoline stimulation did not alter GSH levels in neurons. However, cabergoline concentrations over 10 μM significantly enhanced GSH production in astrocytes (Fig. 1a). Subsequently, GSH levels were measured in astrocytes stimulated with 10 μM cabergoline for different times, and the level of GSH was appreciably increased after 12 h (Fig. 1b). These results indicate that activation of dopamine receptors on astrocytes rather than those on neurons facilitates GSH biosynthesis. Because cabergoline binds with low affinity to dopamine D1 receptor (DRD1) and high affinity to DRD2[37], we pretreated primary astrocytes with the DRD1 antagonist SCH23390 and the DRD2 antagonist sulpiride separately to determine which subtype of dopamine receptor promotes GSH production. Only sulpiride pretreatment diminished cabergoline-induced GSH synthesis (Fig. 1c), indicating that astrocytic DRD2 activation increases GSH levels.

To validate the effects of DRD2 on GSH, we treated astrocytes from wild-type mice and *Drd2*-knockout mice with three selective DRD2 agonists: quinpirole, quinelorane, and bromocriptine. All agonists increased GSH levels in wild-type astrocytes but failed to affect GSH levels in *Drd2*-null astrocytes (Fig. 1d and Supplementary Fig. 1a–c). Next, we crossed *Drd2flox/flox* mice with the human GFAP (hGFAP)-Cre recombinase transgene to generate astrocytic *Drd2* conditional knockout mice. We treated *Drd2flox/flox* and *Drd2flox/flox::hGFAP-Cre* (hereafter referred to as *Drd2hGFAPcKO*) mice with 5 mg kg−1 quinpirole for 7 days. In *Drd2flox/flox* mice, striatal GSH was upregulated after quinpirole administration. In contrast, ablation of *Drd2* did not result in significant alterations in GSH levels (Fig. 1e). These findings demonstrate that astrocytic DRD2 activation promotes GSH synthesis in vitro and in vivo.

As a starting point to identify downstream effectors of DRD2 that might be responsible for regulating GSH production, we employed RNA sequencing to compare the gene transcription profiles of astrocytes isolated from mice after quinpirole administration. The purity of the anti-ACSA-2 microbead kit-isolated astrocytes was verified by flow cytometry (Fig. 1f). Gene set enrichment analysis showed that DRD2 activation increased the targets of Nrf2 (Fig. 1g, h, Supplementary Fig. 2 and Supplementary Data 1), a transcription factor that regulates a series of antioxidant genes[38]. Then, we investigated whether DRD2 activation upregulates Nrf2 targets in primary astrocytes. The expression of most targets of Nrf2, including the catalytic subunit (*Gclc*) and modifier subunit (*Gclm*) of the GSH synthesis rate-limiting enzyme glutamate-cysteine ligase, increased after quinpirole treatment (Fig. 1i and Supplementary Fig. 1d, e). However, quinpirole treatment failed to upregulate Gclc/Gclm expression or GSH biosynthesis in astrocytes from *Nrf2*-knockout mice (Fig. 1j, k), suggesting that DRD2 activation facilitates GSH synthesis through the Nrf2 pathway. Upregulated expression

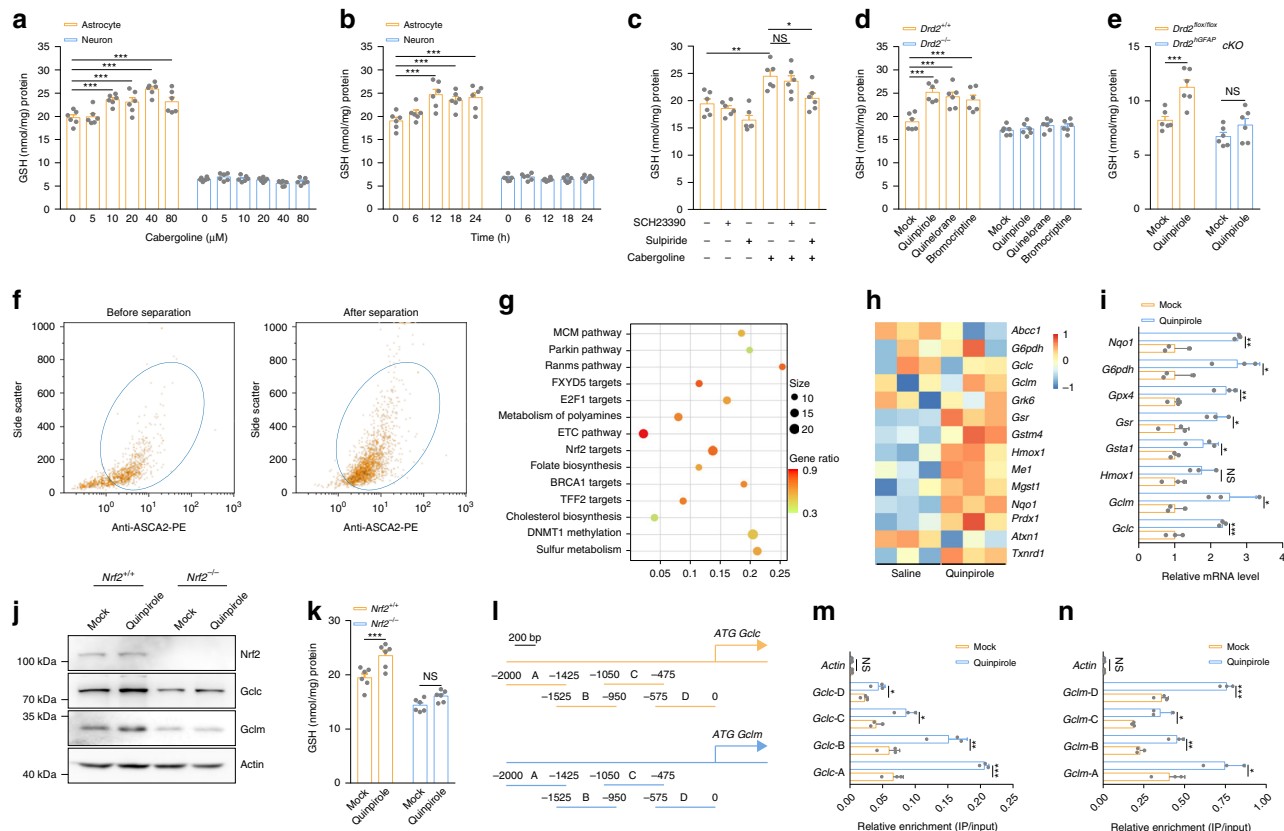

**Fig. 1 Astrocytic DRD2 facilitates GSH synthesis via Nrf2 activation. a** GSH levels in astrocytes and neurons after treatment with 0, 5, 10, 20, 40, or 80 μM cabergoline for 24 h. **b** GSH levels in astrocytes and neurons after treatment with 10 μM cabergoline for 6, 12, 18, or 24 h. **c** GSH levels in astrocytes treated with 10 μM cabergoline for 12 h after pretreatment with SCH23390 (10 μM) or sulpiride (10 μM). **d** GSH levels in astrocytes from wild-type mice or *Drd2*-knockout mice after quinpirole (10 μM), quinelorane (20 μM), or bromocriptine (40 μM) treatment for 12 h. Six independent experiments per condition in **a–d**. **e** Striatal GSH levels in *Drd2*^flox/flox^ mice and *Drd2*^hGFAP^cKO mice after 5 mg kg$^{-1}$ quinpirole administration for 7 days. $n = 6$ mice per group. **f** Representative dot plots of ACSA-2 labeling of astrocytes (gated in blue circles) collected after microbead kit separation. **g, h** Gene set enrichment analysis of upregulated pathways (**g**) and heat map of the expression of Nrf2 targets (**h**) identified by RNA sequencing in astrocytes from mice after quinpirole administration. **i** Astrocytes were treated with 10 μM quinpirole, and Nrf2 target expression was determined by qRT-PCR. Three independent experiments per condition. **j, k** Immunoblotting with actin as a loading control (**j**) and GSH levels (**k**) in wild-type and *Nrf2*-null astrocytes after 10 μM quinpirole treatment. Immunoblotting: three independent experiments; GSH assay: six independent experiments. **l** Diagram of the *Gclc* and *Gclm* promoters showing the locations of the different fragments tested. The numbers indicate the base pairs upstream of the transcriptional start site. **m, n** ChIP assays showing enrichment of the *Gclc* (**m**) and *Gclm* (**n**) promoters in DNA isolated from astrocytes treated with 10 μM quinpirole and precipitated with anti-Nrf2 antibodies. Three independent experiments per condition. Data are presented as the mean ± s.e.m. *$P < 0.05$, **$P < 0.01$, ***$P < 0.001$, NS, not significant. One-way ANOVA with Tukey's multiple comparisons test (**c**), two-way ANOVA with Dunnett's multiple comparisons test (**a**, **b** and **d**) or with Sidak's multiple comparisons test (**e**, **k**), Student's two-tailed unpaired *t*-test (**i**, **m** and **n**). Source data are provided as a Source Data file.

of a transcription factor or enhanced binding to gene promoters enhances the transactivation efficiency. Since DRD2 activation did not alter Nrf2 expression (Fig. 1j), we sought to determine whether DRD2 activation promotes Nrf2 binding to the promoters of *Gclc* and *Gclm*. The results of chromatin immunoprecipitation assays demonstrated that DRD2 activation increases Gclc and Gclm expression by enhancing the binding of Nrf2 to their gene promoters (Fig. 1l–n).

**DRD2 induces dimerization of PKM2 to bind with and activate Nrf2.** Transcription factor binding to gene promoters requires assistance from coactivators[39]. To identify the coactivators responsible for Nrf2 transactivation, we pulled down Nrf2 and used label-free mass spectrometry to detect proteins that bound Nrf2 in greater abundance after DRD2 activation. As shown in Fig. 2a, b, Supplementary Fig. 3, and Supplementary Data 2, pyruvate kinase, which has two isoforms, PKM1 and PKM2, was prominently upregulated after quinpirole treatment. First, via a

coimmunoprecipitation assay, we demonstrated that DRD2 activation enhances the binding of Nrf2 to PKM2 rather than PKM1 (Supplementary Figs. 4a and 5a). To exclude the possibility that the nondetectable level of PKM1 in anti-Nrf2 pulldown products was due to its low expression, we overexpressed PKM1 in astrocytes but still failed to detect its binding with Nrf2 (Supplementary Fig. 4a), confirming that Nrf2 interacts specifically with PKM2. Subsequently, the proximity ligation assay (PLA) results indicated that DRD2 activation induces nuclear-localized PKM2-Nrf2 interactions (Fig. 2c).

Previous studies have reported that PKM2 is enzymatically active toward pyruvate only as a tetramer and acts as a transcription factor coactivator when tetrameric PKM2 dissociates into a dimer[28,29]. The blue native PAGE (BN-PAGE) results showed that DRD2 activation promotes the conversion of PKM2 tetramers to dimers (Fig. 2d). To verify that the PKM2 dimer is a Nrf2 coactivator, we treated astrocytes with DASA-58, a molecule that enhances PKM2 tetramer formation[40]. DASA-58 treatment attenuated DRD2 activation-induced PKM2 dimerization (Fig. 2d)

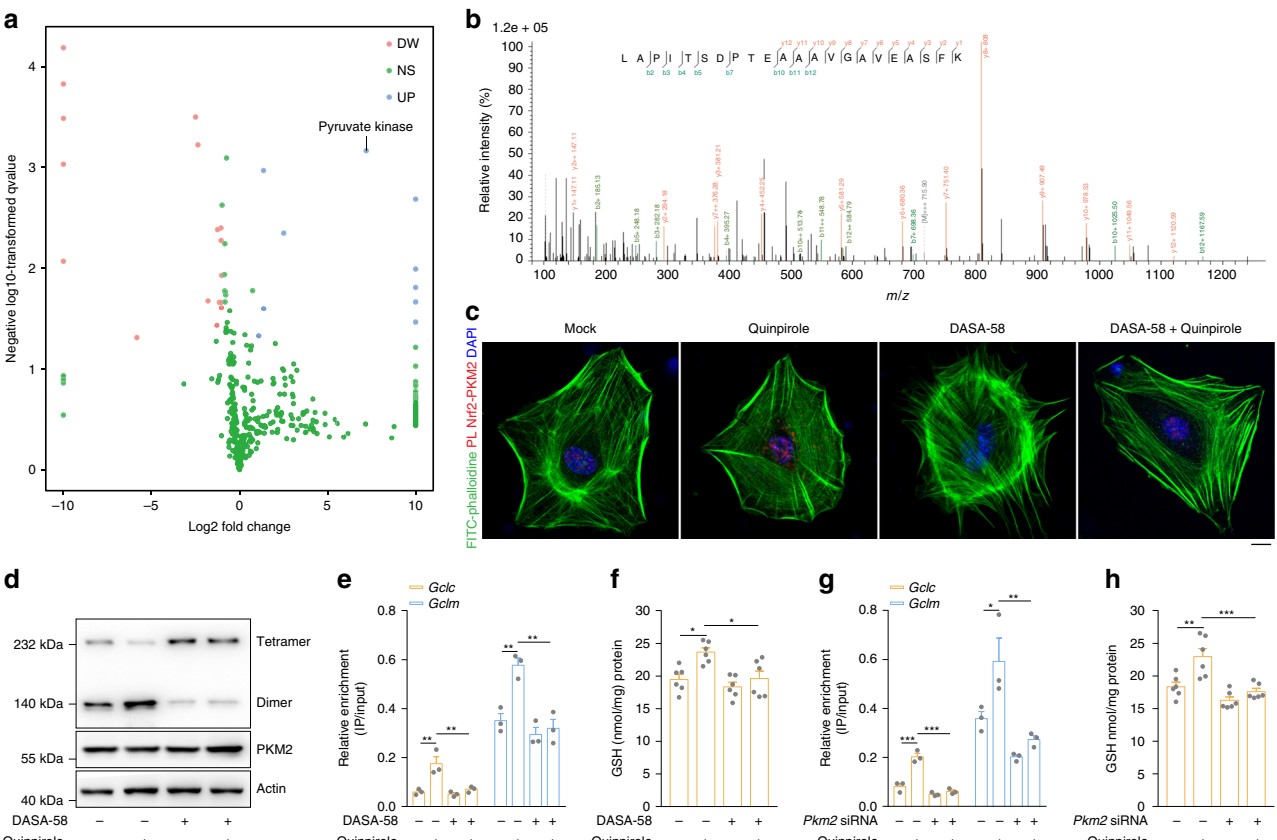

**Fig. 2 DRD2 induces PKM2 dimerization to bind with and activate Nrf2. a** Volcano plot of label-free mass spectrometry-quantified proteins that bound Nrf2 with a significant fold change after DRD2 activation. Only proteins identified in at least two or three replicates with a fold change > 2.0 and a p-value < 0.05 were regarded significantly changed. **b** Representative fragmentation spectrum of [401]LAPITSDPTEATAVGAVEASFK[422] in PKM2. **c**–**f** Primary astrocytes were pretreated with 50 μM DASA-58 for 1 h and then stimulated with 10 μM quinpirole for 12 h. PKM2 and Nrf2 proximity ligation signals (**c**). PKM2 dimer and tetramer formation analyzed by BN-PAGE and total PKM2 detected by immunoblotting with actin as a loading control (**d**). The amount of anti-Nrf2-immunoprecipitated DNA detected by qRT-PCR with primers flanking the Gclc and Gclm promoter regions (**e**) and GSH levels (**f**). Immunoblotting and qRT-PCR: three independent experiments; GSH assay: six independent experiments. Scale bar, 10 μm. **g**, **h** Primary astrocytes transfected with Pkm2-specific siRNA were stimulated with 10 μM quinpirole for 12 h. The amount of anti-Nrf2-immunoprecipitated DNA detected by qRT-PCR with primers flanking the Gclc and Gclm promoter regions (**g**) and GSH levels (**h**). qRT-PCR: three independent experiments; GSH assay: six independent experiments. Data are presented as the mean ± s.e.m. *P < 0.05, **P < 0.01, ***P < 0.001. One-way ANOVA with Tukey's multiple comparisons test (**e**–**h**). Source data are provided as a Source Data file.

and inhibited PKM2 binding to Nrf2 (Fig. 2c and Supplementary Fig. 5a), resulting in decreased Gclc/Gclm expression and GSH synthesis (Fig. 2e, f and Supplementary Fig. 5b, c). In addition, knockdown of PKM2 in astrocytes attenuated DRD2-induced Nrf2 activation and GSH biosynthesis (Fig. 2g, h and Supplementary Fig. 5d–f). These results demonstrate that DRD2 induces dimerization of PKM2 for Nrf2 binding and activation. Moreover, overexpression of the PKM1 isoenzyme did not alter Gclc or Gclm expression (Supplementary Fig. 4b–e), indicating that PKM2 does not affect Nrf2 transactivation through its metabolic function.

**DRD2 activation triggers β-arrestin2 to bind with and dimerize PKM2.** DRD2 activation stimulates both the classical G protein pathway and β-arrestin-biased signaling, resulting in distinct downstream events[41–43]. The Gα$_i$ protein inhibitor pertussis toxin (PTX) did not affect DRD2 activation-induced PKM2 dimerization, suggesting that PKM2 dimerization is independent of the classical G protein pathway (Fig. 3a). Next, β-arrestin1 and β-arrestin2 were knocked down separately (Supplementary Fig. 6a), but only β-arrestin2 knockdown attenuated DRD2 activation-induced PKM2 dimerization (Fig. 3b), indicating that

DRD2 activation facilitates PKM2 dimerization through β-arrestin2. Quinpirole treatment of astrocytes from β-arrestin2-knockout mice failed to induce either PKM2 dimerization or Nrf2 transactivation (Fig. 3c–i), further demonstrating that DRD2 activates the PKM2-Nrf2 pathway through β-arrestin2. Given that β-arrestin2 is a scaffolding protein, we further examined the interaction between β-arrestin2 and PKM2. The coimmunoprecipitation assay results showed that β-arrestin2 bound to PKM2 after DRD2 activation (Supplementary Fig. 6b), suggesting that DRD2 activation drives the interaction of β-arrestin2 with PKM2 to facilitate PKM2 dimerization.

**Divergent expression patterns of PKM in neurons and astrocytes.** These results demonstrate that astrocytic DRD2 induces GSH synthesis via PKM2-mediated Nrf2 transactivation (Supplementary Fig. 7). Our findings indicate that the loss of dopaminergic neurons would start a vicious cycle of decreased dopamine release—DRD2 signaling deficiency—decreased GSH synthesis—oxidative stress—dopaminergic neuron death during the progression of PD and explain the mechanism by which GSH decreases in the brains of PD patients (Supplementary Fig. 8o). According to our results, we can activate astrocytic DRD2, PKM2,

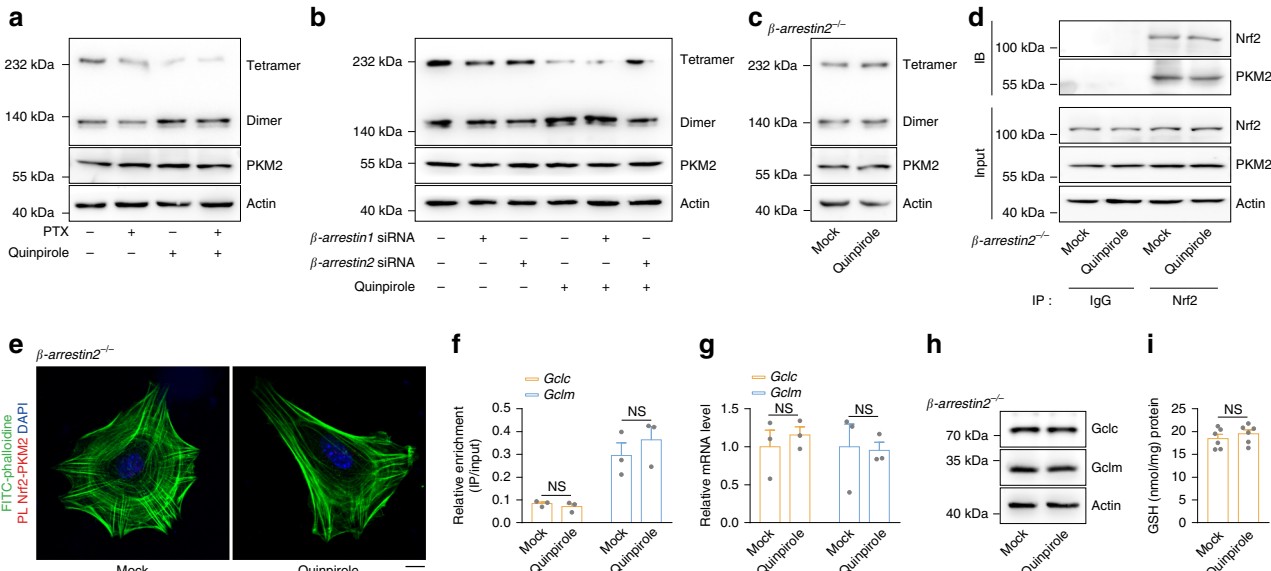

**Fig. 3 DRD2 activation triggers β-arrestin2 to bind and dimerize PKM2. a** The PKM2 dimer and tetramer formation in primary astrocytes pretreated with 100 ng ml$^{-1}$ PTX for 20 min and then stimulated with 10 μM quinpirole for 12 h was analyzed by BN-PAGE, and total PKM2 was detected by immunoblotting with actin as a loading control. Three independent experiments per condition. **b** Primary astrocytes transfected with β-arrestin1- or β-arrestin2-specific siRNA were stimulated with 10 μM quinpirole for 12 h. PKM2 dimer and tetramer formation analyzed by BN-PAGE and total PKM2 detected by immunoblotting with actin as a loading control. Three independent experiments per condition. **c–i** Primary astrocytes from β-arrestin2-knockout mice were stimulated with 10 μM quinpirole for 12 h. PKM2 dimer and tetramer formation analyzed by BN-PAGE and total PKM2 detected by immunoblotting with actin as a loading control (**c**). Immunoblot analysis of PKM2 in cell lysates immunoprecipitated with a Nrf2 antibody (**d**). PKM2 and Nrf2 proximity ligation signals (**e**). Amount of anti-Nrf2-immunoprecipitated DNA detected by qRT-PCR with primers flanking the *Gclc* and *Gclm* promoter regions (**f**). *Gclc* and *Gclm* mRNA detected by qRT-PCR (**g**). Immunoblotting with actin as a loading control (**h**) and GSH levels (**i**). Immunoblotting and qRT-PCR: three independent experiments; GSH assay: six independent experiments. Scale bar, 10 μm. Data are presented as the mean ± s.e.m. NS, not significant. Student's two-tailed unpaired *t*-test (**f**, **g** and **i**). Source data are provided as a Source Data file.

or Nrf2 to recover GSH levels in PD. Accumulating evidence has demonstrated that chemicals targeting DRD2 or Nrf2 increase GSH levels in the brain and confer neuroprotective effects[44–46]. However, because DRD2 and Nrf2 are expressed in both neurons and astrocytes, chemicals targeting either of these targets can affect the physiological functions of neurons and result in unexpected side effects. For example, supplementation with dopamine agonists leads to excessive daytime sleepiness, sleep attacks, dizziness, and hypotension[37,47].

Previous RNA sequencing data analysis showed that neurons and astrocytes express distinct splice isoforms of the *PKM* gene—*PKM1* and *PKM2*, respectively[48]. We validated these data in primary neurons and astrocytes by Q-PCR and immunoblotting (Fig. 4a–d) and in brain slices by multi-immunohistochemical fluorescence (Fig. 4e). These results indicate that astrocytes express mainly PKM2; in contrast, neurons express mainly PKM1, explaining why cabergoline stimulation did not alter GSH levels in neurons but increased GSH synthesis in astrocytes (Fig. 1a, b). Hence, given the low expression of PKM2 in neurons, chemicals targeting PKM2 may not affect the physiological functions of neurons, avoiding unexpected side effects.

**Pyridoxine facilitates GSH synthesis via the PKM2-Nrf2 pathway.** First, we performed an initial screen of 863 natural products purchased from TargetMol by intrinsic tryptophan differential scanning fluorimetry (nanoDSF) to explore molecules that may directly bind with PKM2. According to the shifts in Tm values, we selected 22 molecules based on the highest ΔTm values (Fig. 5a and Supplementary Data 3) and further verified their ability to dimerize PKM2 by BN-PAGE (Fig. 5b). Compound No. 194, vindoline, and Compound No. 324, pyridoxine (also known

as vitamin B$_6$) (Fig. 5c), facilitated PKM2 dimerization in vitro. Second, we employed a high-content screen of those two compounds at different concentrations to observe their ability to improve GSH production and cytotoxicity in primary astrocytes. Pyridoxine had a greater effect on increasing GSH synthesis with lower cytotoxicity than vindoline (Fig. 5d, e). Further studies indicated that pyridoxine treatment of astrocytes resulted in PKM2 dimerization (Fig. 5f) and increased PKM2 binding to Nrf2 (Fig. 5g, h). Pyridoxine treatment also enhanced Nrf2 binding to the promoters of *Gclc/Gclm* (Fig. 5i) and promoted the expression of these genes (Fig. 5j, k), hence increasing GSH biosynthesis (Fig. 5l). However, pyridoxine failed to induce Nrf2 activation and GSH production in PKM2-knockdown astrocytes (Fig. 5i–l) or primary neurons with low PKM2 expression (Fig. 5m, n), suggesting that pyridoxine facilitates GSH synthesis at least partially through the PKM2-Nrf2 pathway.

**Pyridoxine reduces dopaminergic neuron loss in the PD mouse model.** We treated wild-type mice with different concentrations of pyridoxine for 7 consecutive days to determine the effective pyridoxine concentration for inducing GSH synthesis in vivo. As shown in Supplementary Fig. 8a, 5 mg kg$^{-1}$ pyridoxine increased GSH levels as efficiently as quinpirole. In contrast, ablation of *Nrf2* did not result in a significant alteration in GSH levels, indicating that pyridoxine facilitates GSH synthesis at least partially through Nrf2 in vivo (Supplementary Fig. 8b). We further performed BN-PAGE to detect PKM2 dimers and tetramers in the striatum of *Drd2$^{flox/flox}$* mice and *Drd2$^{hGFAP}$cKO* mice following administration of 5 mg kg$^{-1}$ quinpirole or 5 mg kg$^{-1}$ pyridoxine for 7 consecutive days. As shown in Fig. 6a, both quinpirole and pyridoxine promoted PKM2 dimer formation in

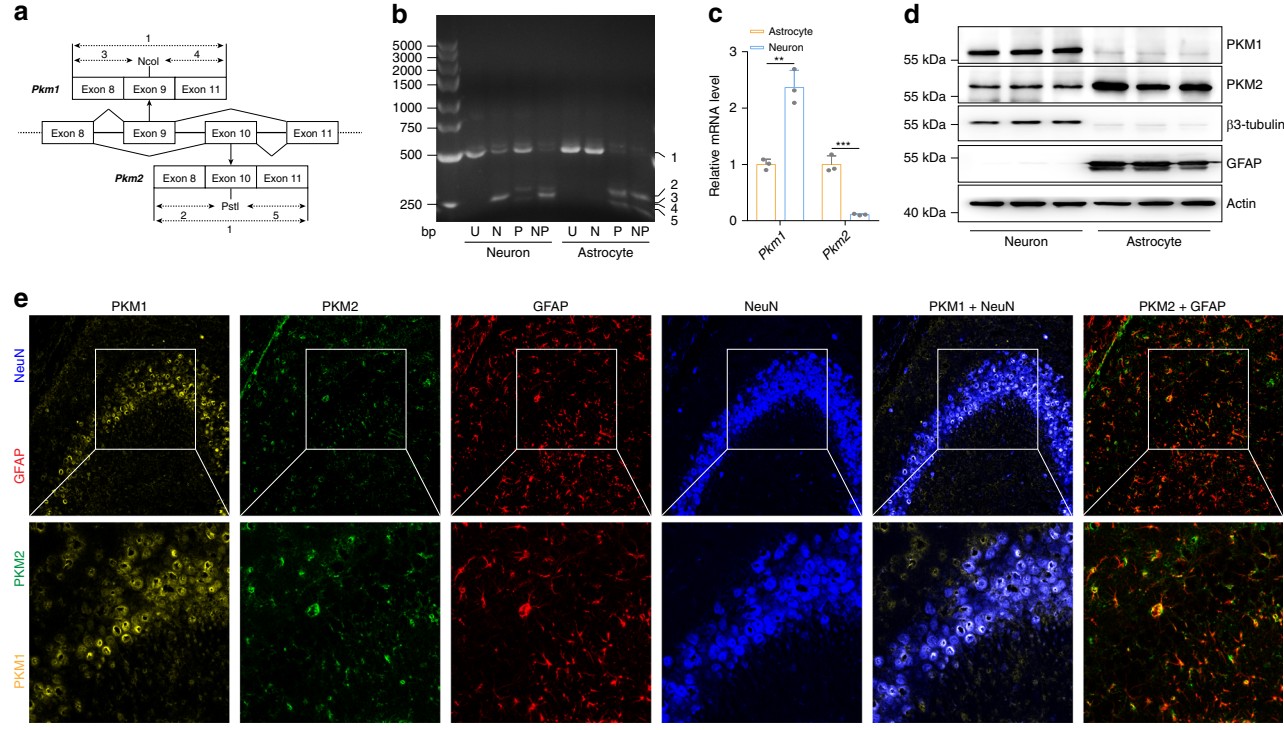

**Fig. 4 Divergent expression patterns of *PKM* in neurons and astrocytes. a** Primers annealing to exons 8 and 11 were used to amplify PKM transcripts. The PCR products were cleaved with Ncol, Pstl or both to distinguish between the PKM1 (including exon 9) and PKM2 (including exon 10) isoforms. **b** RNA from primary neurons and astrocytes was analyzed by qRT-PCR, followed by digestion with Ncol (N), Pstl (P), or both enzymes (NP), plus an uncut control (U). The numbered bands are as follows: 1: uncut M1 or M2 (502 bp); 2: Pstl-cleaved M2 5′ fragment (286 bp); 3: Ncol-cleaved M1 5′ fragment (245 bp); 4: Ncol-cleaved M1 3′ fragment (240 bp); and 5: Pstl-cleaved M2 3′ fragment (216 bp). **c** Pkm1 and Pkm2 mRNA detected by qRT-PCR in primary neurons and astrocytes. **d** Immunoblotting with actin as a loading control. **e** PKM1 and PKM2 immunohistochemical signals in cortical neurons and astrocytes after tyramide signal amplification. Immunoblotting and qRT-PCR: three independent experiments. Scale bar, 50 μm. Data are presented as the mean ± s.e.m. ***$P < 0.001$. Student's two-tailed unpaired *t*-test (**c**). Source data are provided as a Source Data file.

*Drd2^flox/flox* mice. However, only pyridoxine administration resulted in striatal PKM2 dimerization (Fig. 6a) and increased Gclc/Gclm expression (Fig. 6b, c) and GSH biosynthesis (Fig. 6d) in *Drd2^hGFAP^cKO* mice. These results demonstrate pyridoxine promotes GSH synthesis through the PKM2-Nrf2 pathway in a DRD2-independent manner.

Because MPP+ converted from MPTP damages mitochondria and leads to overwhelming ROS production, we selected an MPTP mouse model to assess the effects of GSH induced by pyridoxine on oxidative stress. We pretreated *Drd2^flox/flox* mice and *Drd2^hGFAP^cKO* mice with quinpirole or pyridoxine before MPTP administration (Supplementary Fig. 8c). As shown in Supplementary Fig. 8d, e, neither astrocytic *Drd2* conditional knockout nor pyridoxine treatment altered the toxic MPP+ levels in the striatum. Both quinpirole and pyridoxine administration prevented MPP+-induced loss of nigral dopaminergic neurons (Fig. 6e–h and Supplementary Fig. 8f, g) in *Drd2^flox/flox* mice. In contrast, *Drd2^hGFAP^cKO* mice showed only attenuation of nigral dopaminergic neuron loss induced by MPP+ following pyridoxine treatment but not alleviated neuronal damage following quinpirole treatment (Fig. 6e–h and Supplementary Fig. 8f, g). These results indicate that pyridoxine can reduce the severity of dopaminergic neuron loss when administered before MPTP beyond targeting dopamine receptors.

Motor dysfunction in *Drd2^flox/flox* mice and *Drd2^hGFAP^cKO* mice was evaluated by monitoring voluntary movement behaviors. Pyridoxine treatment shortened the time for turning around (T-Turn) and time for descending a pole (T-TLA) of both *Drd2^flox/flox* mice and *Drd2^hGFAP^cKO* mice in the pole test (Supplementary

Fig. 8h, i) but did not alter the latency time in the rotarod test (Supplementary Fig. 8j) or the activity of mice in the open field test (Supplementary Fig. 8k), indicating that pyridoxine can improve some types of mouse locomotion in a DRD2-independent manner. Since pyridoxine is the coenzyme of some enzymes involved in the synthesis of neurotransmitters, including dopamine and γ-aminobutyric acid (GABA)[49], we further measured striatal dopamine, dihydroxyphenyl acetic acid (DOPAC) and GABA levels after pyridoxine treatment. As shown in Supplementary Fig. 8l–n, pyridoxine treatment slightly increased the DOPAC level but did not alter the dopamine and GABA levels, indicating that supplementation with pyridoxine has few effects on dopamine and GABA levels under physiological conditions.

In conclusion, via experiments in *Nrf2*-knockout mice, we demonstrate that pyridoxine can induce GSH biosynthesis through the PKM2/Nrf2 pathway. Furthermore, by employing the MPTP mouse model to simulate mitochondrial damage and GSH deficiency in PD, we demonstrate that pyridoxine-induced GSH enhancement helps to reduce the severity of dopaminergic neuron loss and improve some types of mouse locomotion after MPTP administration. However, given that pyridoxine is a cocatalyst of many enzymatic processes involved in amino acid metabolisms, such as racemization and decarboxylation, our results do not exclude that pyridoxine may also confer neuroprotection through other undiscovered mechanisms.

## Discussion
In recent years, the nonmetabolic functions of PKM2 in a series of biological processes, such as cell proliferation, apoptosis, and

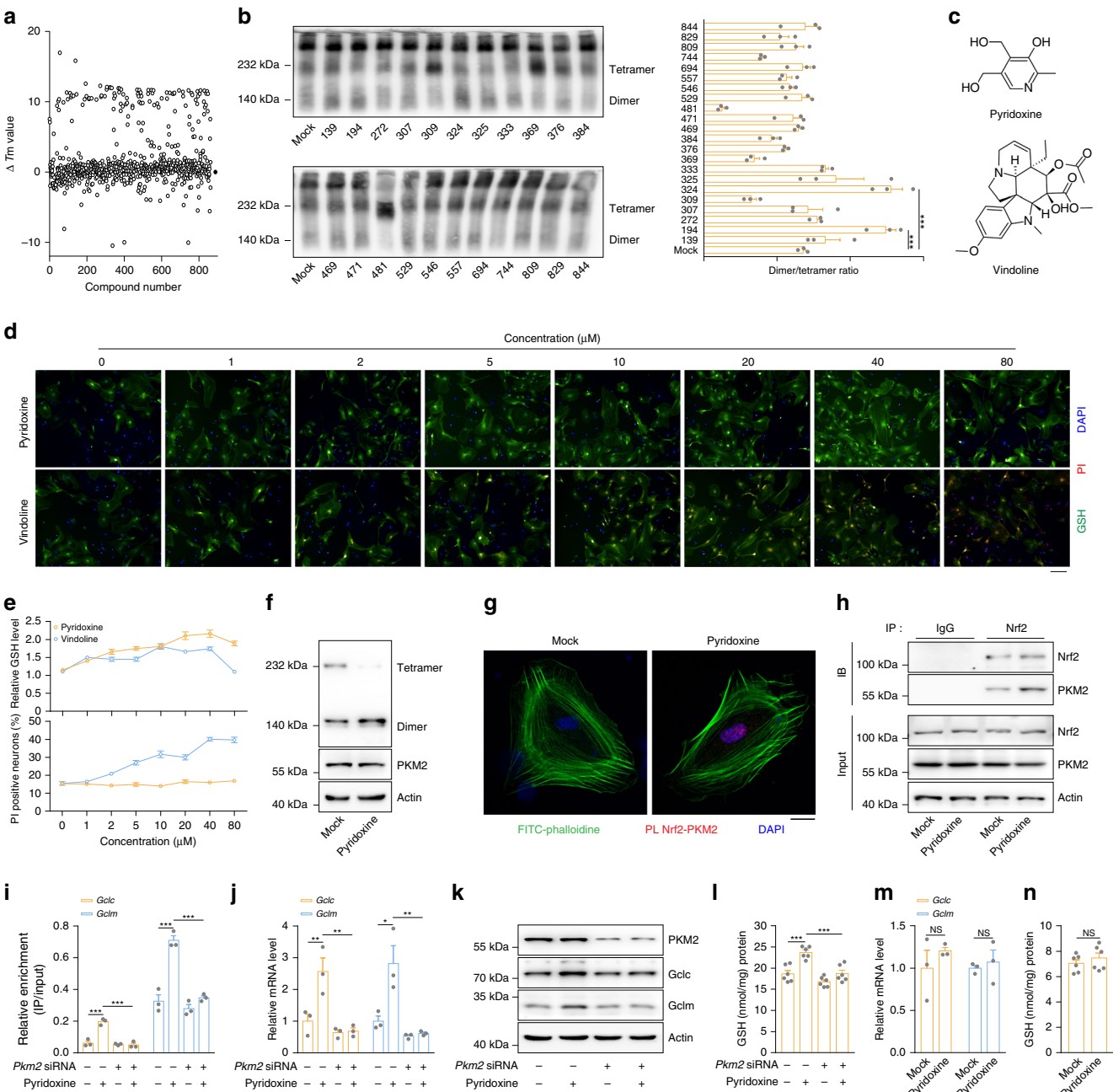

**Fig. 5 Pyridoxine facilitates GSH synthesis via the PKM2-Nrf2 pathway. a, b** Purified recombinant PKM2 (rPKM2) was mixed with 863 natural products individually. ΔTm values analyzed by NanoDSF (**a**). rPKM2 dimer and tetramer formation analyzed by BN-PAGE (left) and densitometric analysis of the dimer/tetramer ratio (right) (**b**). **c** Chemical structures of compound no. 194 (vindoline) and no. 324 (pyridoxine). **d, e** Primary astrocytes were stimulated with different concentrations of pyridoxine or vindoline for 12 h. Immunofluorescence of GSH and PI analyzed by high-content screening (**d**). The relative intensity of GSH signals (upper) and the percentage of PI-positive nuclei (lower) (**e**). Scale bar, 100 μm. **f–h** Primary astrocytes were stimulated with 5 μM pyridoxine for 12 h. PKM2 dimer and tetramer formation analyzed by BN-PAGE and total PKM2 detected by immunoblotting with actin as a loading control (**f**). PKM2 and Nrf2 proximity ligation signals (**g**). Immunoblot analysis of PKM2 in cell lysates immunoprecipitated with a Nrf2 antibody (**h**). Immunoblotting: three independent experiments. Scale bar, 10 μm. **i–l** Primary astrocytes transfected with *Pkm2*-specific siRNA were stimulated with 5 μM pyridoxine for 12 h. The amount of anti-Nrf2-immunoprecipitated DNA analyzed by qRT-PCR with primers flanking the *Gclc* and *Gclm* promoter regions (**i**). *Gclc* and *Gclm* mRNA detected by qRT-PCR (**j**). Immunoblotting with actin as a loading control (**k**) and GSH levels (**l**). Immunoblotting and qRT-PCR: three independent experiments; GSH assay: six independent experiments. **m, n** Primary neurons were stimulated with 5 μM pyridoxine for 12 h. *Gclc* and *Gclm* mRNA detected by qRT-PCR (**m**) and GSH levels (**n**). qRT-PCR: three independent experiments; GSH assay: six independent experiments. Data are presented as the mean ± s.e.m. *$P < 0.05$, **$P < 0.01$, ***$P < 0.001$, NS, not significant. One-way ANOVA with Dunnett's multiple comparisons test (**b**) or with Tukey's multiple comparisons test (**i, j** and **l**). Student's two-tailed unpaired *t*-test (**m, n**). Source data are provided as a Source Data file.

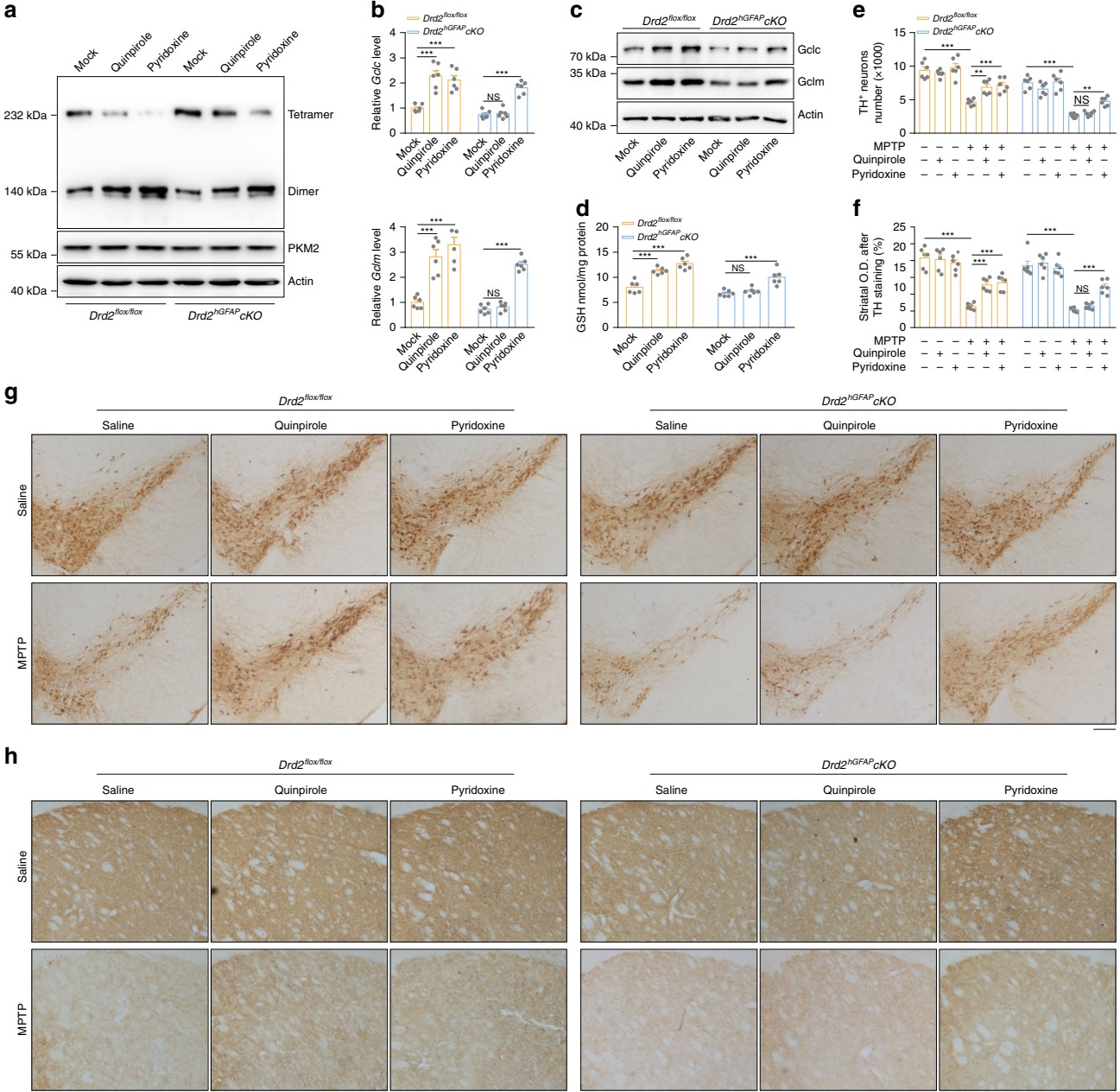

**Fig. 6 Pyridoxine reduces dopaminergic neuron loss in the PD mouse model. a–d** *Drd2*$^{flox/flox}$ and *Drd2*$^{hGFAP}$*cKO* mice were sacrificed after continuous infusion of 5 mg kg$^{-1}$ quinpirole or 5 mg kg$^{-1}$ pyridoxine for 7 consecutive days, and the striatum was dissected for further analysis. PKM2 dimer and tetramer formation analyzed by BN-PAGE and total PKM2 detected by immunoblotting with actin as a loading control (**a**). *Gclc* and *Gclm* mRNA detected by qRT-PCR (**b**). Immunoblotting with actin as a loading control (**c**) and GSH levels (**d**). Immunoblotting and qRT-PCR: three independent experiments; GSH assay: six independent experiments. **e–h** *Drd2*$^{flox/flox}$ and *Drd2*$^{hGFAP}$*cKO* mice were used to generate an MPTP-induced PD mouse model (20 mg kg$^{-1}$ i.h., 5 **d**), which was treated by continuous infusion of quinpirole (5 mg kg$^{-1}$ i.p.) or pyridoxine (5 mg kg$^{-1}$ i.p.). Quantification of TH$^+$ neurons (**e**). Striatal optical density (O.D.) measurements of TH-stained slices (**f**). Immunohistochemical staining of TH$^+$ neurons (**g**). Immunohistochemical staining of TH in the nigrostriatal system (**h**). n = 6 mice per group. Scale bar, 200 μm. Data are presented as the mean ± s.e.m. **P < 0.01, ***P < 0.001, NS, not significant. Two-way ANOVA with Dunnett's multiple comparisons test (**b**, **d**) or with Tukey's multiple comparisons test (**e**, **f**). Source data are provided as a Source Data file.

cytokinesis, have sparked considerable interest[50–54]. Our results uncover a new noncanonical function of PKM2 as a transcriptional coactivator of Nrf2, which cooperates with its metabolic function to divert glucose flux into the pentose phosphate pathway[55] and thereby generates sufficient GSH for ROS detoxification. Moreover, our results indicate that tumor cells can employ PKM2 to produce GSH and escape from oxidative stress, which

explains the essentiality of PKM2 for tumor proliferation[50] from another perspective.

Existing therapeutic strategies for PD treatment, which attempt to compensate for the loss of dopamine, are limited by their side effects and lack of long-term efficacy[47]. Hence, understanding dopamine receptor signaling and avoiding the activation of downstream events that produce side effects is an approach for

improving PD therapies. Auspiciously, we found that PKM2 mediates DRD2-induced GSH production and is selectively expressed in astrocytes, indicating that PKM2 dimerization is an ideal approach to restore GSH synthesis and prevent side effects induced by effects on neuron functions. These advantages make PKM2 a potential target for PD treatment.

Our results demonstrate that pyridoxine, also known as vitamin B$_6$, induces GSH synthesis via PKM2-mediated Nrf2 transactivation and confers neuroprotection. Notably, pyridoxine has an interaction with levodopa (L-DOPA), the mainstay in PD treatment. On the one hand, pyridoxine is consumed by L-DOPA metabolism; thus, the use of levodopa can cause symptomatic pyridoxine deficiency[56,57]. On the other hand, pyridoxine is the coenzyme of dopa decarboxylase, converting L-DOPA to dopamine[49], which raises a concern that pyridoxine supplementation may result in peripheral metabolism of L-DOPA and a lack of bioavailability in the brain[58]. However, recent studies have demonstrated that a pyridoxine intake of at most 50 mg day$^{-1}$ would not reduce the efficacy of L-DOPA[59]. In summary, proper supplementation of pyridoxine is essential to L-DOPA treatment and does not reduce the efficacy of L-DOPA.

The current recommended dietary allowance (RDA) of pyridoxine according to the American National Institute of Health (NIH) is 2 mg day$^{-1}$, with an upward tolerance of 100 mg day$^{-1}$ for adults[49]. Though consecutive high-dose pyridoxine intake can lead to peripheral sensory neuropathy and nerve degeneration, these problems are generally reversible when supplementation is stopped[60,61]. Moreover, in our animal models, pyridoxine treatment did not induce notable weight loss or movement disorders. Thus, we postulate that pyridoxine is a relatively safe candidate for PD therapy. Interestingly, consistent with our findings, several epidemiological studies have reported that dietary pyridoxine significantly decreases the risk of PD[62-64].

The present study, summarized in Supplementary Fig. 7, illustrates the molecular mechanism by which astrocytic DRD2 induces GSH biosynthesis. These findings suggest that after the loss of dopaminergic neurons, a vicious cycle of decreased dopamine release—DRD2 signaling deficiency—decreased GSH synthesis—oxidative stress—dopaminergic neuron death will occur, leading to GSH deficiency and the progression of PD. Moreover, given that dopamine agonist treatment in PD is limited by their intolerable side effects and diminished effectiveness over time, we selected astrocyte-specific PKM2 as a target to break this vicious cycle and found that pyridoxine can dimerize PKM2 to restore GSH levels, providing neuroprotection in the MPTP mouse model beyond targeting dopamine receptors. Although more PD models are needed to validate whether PKM2 is a universal therapeutic target in PD, our results provide preliminary evidence that dimerizing PKM2 is a potential therapeutic approach to protect nigral dopaminergic neurons from oxidative damage.

## Methods

**Mice**. All animal care and procedures were performed following national and international guidelines and were approved by the Animal Resource Centre, Nanjing Medical University. We complied with all relevant ethical regulations for animal testing and research. Mice were housed in groups (2–5 siblings) at 22–24 °C with a 12 h light-dark cycle and ad libitum access to regular chow diet and water. Adult or neonatal C57BL/6 mice were purchased from the Model Animal Research Center of Nanjing University. Drd2-knockout mice on a C57BL/6 (inbred) genetic background were generated by 10 backcrosses. Drd2-floxed mice were created by Shanghai Research Center for Model Organisms. hGFAP-Cre transgenic mice on a C57BL/6 genetic background, which were initially derived from GFAP-Cre mice (Jax Stock No: 004600), were a gift from Jiawei Zhou. Nrf2-knockout mice (Jax Stock No: 017009) were a gift from Peng Cao. β-Arrestin2-knockout mice (Jax Stock No: 011130) were obtained from Gang Pei.

Three-month-old Drd2$^{flox/flox}$ and Drd2$^{hGFAP}$ cKO mice were used to generate a subacute MPTP-induced PD model. They were injected hypodermically (i.h.) with 20 mg kg$^{-1}$ MPTP (Sigma-Aldrich, M0896) dissolved in saline once a day for 5 consecutive days and then left alone for 3 days. Mice in the treatment groups

received 5 mg kg$^{-1}$ quinpirole (Sigma-Aldrich, Q111) or 5 mg kg$^{-1}$ pyridoxine (HY-B1328) intraperitoneally (i.p.) daily starting 1 day before MPTP administration and continuing over the 5 MPTP injection days and the 3 subsequent days. Three days after the last MPTP injection, the mouse striatum was dissected and processed for immunoblotting, qRT-PCR, or GSH assay. In some cases, animals were perfused with 4% paraformaldehyde in 0.1 M phosphate buffer (pH 7.4), and coronal cryo-sections at a thickness of 25 mm were prepared for immunohistochemistry.

**Reagents**. Cabergoline (HY-15296), bromocriptine (HY-12705A), SCH23390 (HY-19545A), DASA-58 (HY-19330), and pyridoxine (HY-B1328) were purchased from MedChemExpress. Quinpirole (Q111) and PTX (SAE0066) were purchased from Sigma-Aldrich. Sulpiride (S4655) and vindoline (S3970) were purchased from Selleck. Quinelorane (1519) was purchased from TOCRIS. The natural products library was purchased from TargetMol (Shanghai, China). FITC-phalloidine (YP0059) was purchased from US Everbright®Inc.

**Cell culture and transfection**. Brain tissue was isolated from day 16 C57BL/6J embryos. The meninges and basal ganglia were removed under a microscope. Tissues were digested with 0.025% trypsin at 37 °C for 30 min; the reaction was terminated with Dulbecco's modified Eagle's medium (DMEM, Gibco, 12100-046) supplemented with 10% fetal bovine serum (Gibco, 10437028). The suspension was filtered with a 40 μm filter (BD Falcon, 352340) and centrifuged at 300 × g for 5 min. The cell precipitate was resuspended in Neurobasal medium (Gibco, 21103049) supplemented with 2% B-27 (Gibco, 17504044) and 0.5 mM L-glutamine, and the cells were seeded on plates pre-coated with poly-D-lysine hydrobromide (Sigma-Aldrich, P0296). Neurons were cultured for 7 days, and half of the medium was replaced with fresh medium every 3.5 days.

Primary astrocyte was isolated from brain tissue of neonatal C57BL/6J mice aged 1–3 days. The meninges and basal ganglia were removed under a microscope and digested with 0.25% trypsin; the reaction was terminated with FBS. The suspension was filtered with a 40 μm filter (BD Falcon, 352340) and centrifuged at 300 × g for 5 min. The cell precipitate was resuspended in DMEM supplemented with 10% FBS and 1% penicillin-streptomycin (Gibco, 15640055), and the cells were seeded. The medium was replaced with fresh medium 12–18 h later and changed every 3 days. DMEM medium was replaced by Medium 199 (Gibco, 11150067) before pyridoxine treatment. The β-arrestin1 siRNA (5′-CCAGCUCA ACAUUCUGCAAUU-3′) and β-arrestin2 siRNA (5′-GCUUGUGGAGUAGAC UUUGUU-3′) purchased from GenePharma (Shanghai, China) were transfected into astrocytes using Lipofectamine 3000 reagent (Invitrogen, L3000015) according to the manufacturer's protocol. siRNA targeting Pkm2 (5′-CTTGCAGCTATT CGAGGGAA-3′) was designed by and purchased from RiboBio (Guangzhou, China). A control siRNA was used as a negative control. Adenovirus overexpressing 3 X -FLAG-tagged PKM1 for introduction into primary astrocytes was purchased from Hanbio Biotechnology (Shanghai, China).

**Total GSH measurement**. Total glutathione levels were measured according to the manufacturer's protocol by using the Glutathione Assay Kit (703002) from Cayman Chemical. Briefly, primary cells were washed with PBS, lysed in 100 μl of 0.4 N perchloric acid (PCA) for 30 min at 4 °C, and centrifuged, and the supernatants were neutralized with four volumes of 0.1 M NaH$_2$PO$_4$ and 5 mM EDTA, pH 7.5. GSH content was measured in a P96 automatic reader by the addition of 5,5′ dithio-bis-2-nitrobenzoic acid (0.6 mM), NADPH (0.2 mM), and glutathione reductase (1 U), and the reaction was monitored at 412 nm for 6 min.

**Adult mouse astrocyte isolation**. Brain tissues were aseptically removed, minced and enzymatically digested in HBSS (Gibco, 14175103) containing 50 mg ml$^{-1}$ collagenase D (Sigma-Aldrich, 11088858001), 100 μg ml$^{-1}$ TLCK trypsin inhibitor (Sigma-Aldrich, T9378), 100 U μl$^{-1}$ DNase I (Sigma-Aldrich, D5025), and HEPES pH 7.2 (Gibco, 15630080) for 1 h at 23 °C with shaking. The tissue was pushed through a 70 μm filter and centrifuged at 500 × g for 10 min. The cell pellet was resuspended in a 37% Percoll solution and centrifuged at 1200 × g for 30 min to remove myelin debris. The cells were washed in PBS and then resuspended in MACS buffer. ACSA-2$^+$ astrocytes (Miltenyi Biotec, 130-097-678) were isolated using manual MACS according to the manufacturer's instructions.

**RNA isolation and qRT-PCR**. Total RNA was extracted from primary astrocytes and striatum tissue using TRIzol reagent (Invitrogen, 15596026) and reversed transcribed into cDNA using PrimeScript™ RT Master Mix (Takara, RR036A). Real-time PCR was performed in a 10 μl reaction system containing cDNA, primers, and SYBR (Roche, 04913914001) with the ABI system. Actin was used as an internal control gene. qPCR primers were designed using a primer design tool, and their sequences were as follows: Pkm1, forward 5′-CACCGTCTGCTGTTTGAAG A-3′, reverse 5′-TCAAAGCTGCTGCTAAACACTT-3′; Pkm2, forward 5′-AGGC TGCCATCTACCACTTG-3′, reverse 5′-CACTGCAGCACTTGAAGGAG-3′; Nqo1, forward 5′-AGGGCAGAAGGGAATTGCTC-3′, reverse 5′-AAAGAGCT GGAGAGCCAACC-3′; G6pdh, forward 5′-CCACTCCAGAAGAAAGACCTA AG-3′, reverse 5′-TGGCTGTTGAGGTGCTTATAG-3′; Gpx4, forward 5′-GCCAA AGTCCTAGGAAACGC-3′, reverse 5′-CCGGGGTTGAAAGGGTTCAGGA-3′; Gsr1,

forward 5′-GCCGCCTGAACACCATCTAT-3′, reverse 5′-CGAGGACCATCT GCGAATGT-3′; *Gsta1*, forward 5′-CCGTTACTTGCCTGCCTTTG-3′, reverse 5′-ATTGGGGAGGCTGCTGATTC-3′; *Hmox1*, forward 5′-GCTGCTCGCTCAC GGT-3′, reverse 5′-GATTCAGGCTCCGGGCTATG-3′; *Gclc*, forward 5′-GAT TCAGGCTCCGGGCTATG-3′, reverse 5′-TGGGTCTCTTCCCAGCTCAGT-3′; *Gclm*, forward 5′-TTGCTGCTCAGTTGGACTCA-3′, reverse 5′-TTGCTGCTCA GTTGGACTCA-3′; and *Actin*, forward 5′-GGACTGTTACTGAGCTGCGTT-3′, reverse 5′-CGCCTTCACCGTTCCAGTT-3′. Following PCR amplification, a first derivative melting curve analysis was performed to confirm the specificity of the PCR. The relative fold difference in mRNA between samples was calculated by comparing the threshold cycle (Ct) at which product initially appeared above background according to $2^{-(\Delta Ct)}$, where $\Delta Ct$ is the difference between the control group and treatment group.

**RNA sequencing**. Sequencing libraries were generated using the NEBNext®Ul-traTM RNA Library Prep Kit for Illumina following the manufacturer's recommendations, and index codes were added to attribute sequences to each sample. Clustering of the index-coded samples was performed on a cBot Cluster Generation System using the TruSeq PE Cluster Kit v4-cBot-HS (Illumina) according to the manufacturer's instructions. After cluster generation, the libraries were sequenced on an Illumina HiSeq 2500 platform, and paired-end reads were generated. The adaptor sequences and low-quality sequence reads were removed from the data sets. Raw sequences were transformed into clean reads after data processing. These clean reads were then mapped to the reference genome sequence. Only reads with a perfect match or one mismatch were further analyzed and annotated based on the reference genome. TopHat2 tools were used to map sequences to the reference genome. The RNA sequencing data that support the findings of this study have been submitted to the NCBI Gene Expression Omnibus under the accession code GSE143164.

**Coimmunoprecipitation (co-IP) assay and immunoblotting**. Striatum tissues and cells were lysed using RIPA lysis buffer (20 mM Tris-HCl, 150 mM NaCl, 1% Triton X-100, and 0.5% NP-40) containing protease and phosphatase inhibitors for 1 h on ice and sonicated 3 times using a Sonics VC750 ultrasonic processor. Then, extracted proteins were quantitated by the BCA method and denatured by boiling at 95 °C for 5 min. Protein samples were separated by sodium dodecyl sulfate-polyacrylamide gel electrophoresis (SDS-PAGE) and transferred to hydrophobic polyvinylidene (PVDF) membranes, which were blocked with 10% milk for 1 h at room temperature. Then, the membranes were incubated at 4 °C overnight with primary antibodies against PKM1 (1:1000, Cell Signaling Technology, 7067), PKM2 (1:1000, Cell Signaling Technology, 4053), Nrf2 (1:1000, Cell Signaling Technology, 12721), β-arrestin1 (1:1000, Cell Signaling Technology, 12697), β-arrestin2 (1:1000, Cell Signaling Technology, 3857), β3-tubulin (1:1000, Cell Signaling Technology, 4466), β-actin (1:1000, Cell Signaling Technology, 4970), Gclc (1:1000, Proteintech, 12601-1-AP), Gclm (1:1000, Proteintech, 14241-1-AP), and GFAP (1:1000, Millipore, MAB360). Then, the membranes were incubated for 1.5 h at room temperature with goat anti-rabbit IgG (H + L) secondary antibody, HRP (Invitrogen, 65-6120). Bands were detected by Novex™ ECL Chemiluminescent Substrate Reagent Kit (Invitrogen, WP20005) using an ImageQuant LAS 4000 mini (GE Healthcare).

For co-IP, proteins from primary astrocytes were harvested by lysis in cell lysis buffer (150 mM NaCl, 0.5% NP-40, 10% glycerol, 2 mM mercaptoethanol, 0.2 mM phenylmethylsulfonyl fluoride, and 10 mM glycerophosphate) after experimental treatments. Lysates were incubated with the anti-Nrf2 antibody (Proteintech, 66504-1-lg), anti-β-arrestin2 antibody (Santa Cruz, sc-514791), or mouse IgG conjugated to sepharose beads (Cell Signaling Technology, 3420). Immunoprecipitates were pelleted, washed twice with cell lysis buffer, washed with PBS and then eluted in 100 mM glycine-HCl, pH 2.5, by incubation for 1 h on ice. The eluate was neutralized by the addition of 1 M Tris-HCl, pH 8.0. The eluate was analyzed by immunoblotting.

**Chromatin immunoprecipitation (ChIP)**. ChIP assays were performed using a ChIP Assay Kit (Millipore, 17-295). Primary astrocytes were crosslinked with 1% formaldehyde for 20 min at 37 °C and quenched in 0.125 M glycine. DNA was immunoprecipitated from the sonicated cell lysates with the anti-Nrf2 antibody (Cell Signaling Technology, 12721) and quantified using qRT-PCR. Primer sequences are as follows:

*Gclc*-A, forward 5′-GCGCACACACACACACACAC-3′, reverse 5′-TCCTCCC CACCATCCTTGCT-3′; *Gclc*-B, forward 5′-GCCCCAAGCTCTTTGATGGC-3′, reverse 5′-GCCCAACCTGGCCTTGAACT-3′; *Gclc*-C, forward 5′-AGCCTTC CGCACTCAGGGTA-3′, reverse 5′-TGGGATCTCTGGTCGGCACA-3′; *Gclc*-D, forward 5′-AGGACAGCCAGGGCTACACA-3′, reverse 5′- GCGGGGCGTGTCT GCGAATAA-3′; *Gclm*-A, forward 5′-GAGGAAGCAAGGCACCTCCC-3′, reverse 5′- AGGTGGGCGGAGGAACCTAA-3′; *Gclm*-B, forward 5′-CACTCCTGAGTGT GCCAGCC-3′, reverse 5′-TTATGGGCACCGTTCAGGGC-3′; *Gclm*-C, forward 5′- GGCTGCTCCTGGGAAACCTC-3′, reverse 5′-TCCCACTGGAGATGGCT GGG-3′; *Gclm*-D, forward 5′-AAGCAGGCACGCTCTCAAGG-3′, reverse 5′-TCG TGAGAGTTGACGGTGCG-3′.

**Label-free mass spectrometry**. Anti-Nrf2 immunoprecipitated protein (250 μg for each sample) was digested according to the FASP procedure. Briefly, the detergent, DTT and other low-molecular-weight components were removed using 200 μl UA buffer by repeated ultrafiltration facilitated by centrifugation. The protein suspension was digested with 3 μg trypsin in 40 μl 25 mM NH$_4$HCO$_3$ overnight at 37 °C. After digestion, the peptides in each sample were desalted on C18 cartridges, concentrated by vacuum centrifugation and reconstituted in 40 μl of 0.1% (v/v) trifluoroacetic acid. MS experiments were performed on a Q Exactive mass spectrometer coupled to an Easy nLC. Five micrograms of the peptide were loaded onto a C18-reversed-phase column in buffer A (2% acetonitrile and 0.1% formic acid) and separated with a linear gradient of buffer B (80% acetonitrile and 0.1% formic acid) at a flow rate of 250 nL min$^{-1}$ controlled by IntelliFlow technology over 120 min. MS data were acquired using a data-dependent top10 method, which dynamically chooses the most abundant precursor ions from the survey scan (300–1800 m z$^{-1}$) for HCD fragmentation. Determination of the target value is based on predictive automatic gain control. Survey scans were acquired at a resolution of 70,000 at m z$^{-1}$ 200, and the resolution for HCD spectra was set to 17,500 at m z$^{-1}$ 200. The instrument was run with peptide recognition mode enabled. MS experiments were performed in triplicate for each sample.

MS data were analyzed using MaxQuant software version 1.3.0.5 and were searched against the UniProtKB database. An initial search was set at a precursor mass window of 6 ppm. The search followed an enzymatic cleavage rule of trypsin and allowed a maximum of two missed cleavage sites and a mass tolerance of 20 ppm for fragment ions. Carbamidomethylation of cysteines was defined as a fixed modification, while protein N-terminal acetylation and methionine oxidation were defined as variable modifications for database searching. The cutoff for the global false discovery rate (FDR) for peptide and protein identification was set to 0.01. Label-free quantification was carried out in MaxQuant. Protein abundance was calculated by the normalized spectral protein intensity (LFQ intensity).

**Proximity ligation assay**. Protein interactions in astrocytes were detected using the Duolink® PLA assay kit (Sigma-Aldrich, DUO92101) following the manufacturer's protocol. After treatments, cells on slides were fixed with 4% paraformaldehyde and permeabilized with 0.3% Triton X-100 in PBS. Blocking solution was added to the slides, which were incubated at 37 °C for 1 h. Then, the slides were incubated with mouse primary Nrf2 antibody (Proteintech, 66504-1-lg) and rabbit primary PKM2 antibody (Cell Signaling Technology, 4053) at 4 °C overnight and then with PLA probe solution for 1 h at 37 °C. After being washed, the slides were incubated for 30 min at 37 °C and then incubated with the amplification solution at 37 °C for 100 min protected from light. Finally, cell nuclei were stained with DAPI (Invitrogen, D1306), and the slides were imaged using the confocal laser scanning microscopy platform Leica TCS SP8.

**Blue native PAGE (BN-PAGE) analysis**. Dimeric and tetrameric PKM2 was detected by BN-PAGE. Briefly, total protein was extracted from primary astrocytes using BN-PAGE lysis buffer (50 mM Bis Tris-HCl, 0.5 M 6-amino-caproic acid, 10% glycerol, and 1% digitonin) containing protease and phosphatase inhibitors. After lysis on ice for 30 min, the sample was centrifuged at 16,000 × g for 10 min, and the supernatant was retained and quantitated. BN-PAGE sample buffer was added to the samples, which were loaded on 8% BN-PAGE gels. Electrophoresis was performed at 4 °C and 100 V using anode and cathode buffers, followed by protein transfer onto PVDF membranes and methanol decolorization. The following steps were the same as those for immunoblotting.

**Multiplex tyramide signal amplification**. Brain slices were treated as described for immunohistochemistry in terms of the quenching of endogenous peroxidase activity, permeabilization, and blocking. Antigens were successively detected using the following protocol. Briefly, each primary antibody was incubated for 12 h in a humidified chamber at 37 °C, followed by detection using the HRP-conjugated secondary antibody and TSA-dendron-fluorophores (Histova, NEON 7-color IHC Kit for cryosection, 1:100), after which the primary and secondary antibodies were thoroughly eluted in stripping solution (Histova, Abcracker) for 60 min at 37 °C. In a serial fashion, each antigen was labeled by distinct fluorophores. After all the antibodies were detected sequentially, the slices were finally stained with DAPI. Multiplex TSA-stained brain slices were imaged using the confocal laser scanning microscopy platform Leica TCS SP8.

**Pyruvate kinase mRNA splicing assay**. Two microgram of total RNA was extracted from primary neurons or astrocytes using Trizol reagent. Contaminating DNA was removed by treatment with DNase I. Reverse transcription was carried out using ImPromp-II reverse transcriptase. After 35 amplification cycles, the amplification products were divided into four aliquots for digestion with NcoI (New England Biolabs, NEB-R3193S), PstI (New England Biolabs, NEB-R3140S), both enzymes, or neither enzyme at 37 °C for 30 min. The digestion products were separated in a 1.5% agarose gel and visualized by a gel imaging system (Tanon 3500).

**Thermal unfolding assay**. Purified recombinant PKM2 (rPKM2) was used for thermal unfolding experiments. The protein was diluted in 20 mM HEPES,

150 mM NaCl, 10% glycerol to a final concentration of 0.2 mg ml$^{-1}$. Samples were supplemented with the indicated natural products to a final concentration of 50 μM and then incubated for 30 min on ice before being loaded into UV capillaries, and experiments were carried out using Prometheus NT.48 for nanoDSF (NanoTemper Technologies). The temperature gradient was set to +1 °C min$^{-1}$ from 20 °C to 90 °C. Protein unfolding was measured by detecting the temperature-dependent change in tryptophan fluorescence at emission wavelengths of 330 and 350 nm. ΔTm shift values were obtained for the 863 natural products, and the 22 molecules with the highest ΔTm values were selected for further analysis.

**Immunohistochemistry analysis**. Mice in different treatment groups were anesthetized with pentobarbital sodium (Sigma-Aldrich, P-010) and perfused with PBS followed by 4% paraformaldehyde in PBS. The brains were harvested and post-fixed in 4% paraformaldehyde, followed by dehydration in 20% sucrose-PBS and then 30% sucrose-PBS. Next, the brains were cut into frozen slices (25 μm). After being rinsed with PBS, the sections were treated with 3% H$_2$O$_2$ for 15 min to quench endogenous peroxidase activity, permeabilized and blocked with 0.3% Triton X-100 in PBS containing 5% BSA for 1.5 h at room temperature. Then, the slices were incubated with primary TH antibody (Sigma-Aldrich, T1299) and primary GFAP antibody (Millipore, MAB360) at 4 °C overnight and then with the corresponding secondary antibody, and the results were visualized by the DAB reaction. For Nissl staining, slices were stained in 0.1% cresyl violet acetate solution (0.1 g cresyl violet in 99% H$_2$O and 1% acetic acid) for 30 min at room temperature. Then, slices were dehydrated with ethanol and xylene. Stereo Investigator software was used to image and count the number of positive cells under the microscope (Olympus BX51).

**Ultra high-performance liquid chromatography (UHPLC)**. MPP$^+$, monoamines and amino acids were measured using UHPLC. Blood plasma samples were collected in heparin (100 U ml$^{-1}$) pretreated tube and centrifuged at 1500 × g for 10 min. The supernatant was mixed with 2 M perchloric acid (10 μl supernatant in 100 μl perchloric acid). The mixture was vortexed immediately, treated by ultrasonic and centrifuged at 20,000 × g for 30 min. Then the supernatant was collected for measuring. Tissue samples were homogenized with 0.1 M perchloric acid (1 mg tissue in 100 μl perchloric acid), treated by ultrasonic and centrifuged at 20,000 × g for 30 min. Then the supernatant was collected for measuring. Both blood plasma and tissue samples were analyzed by Thermo Fisher UHPLC-UV system with Hypersil GOLD VANQUISH C18 UHPLC columns. For MPP$^+$ quantification, the mobile phase is KH$_2$PO4 (50 mmol l$^{-1}$, pH 3.2): acetonitrile = 85:15 with flow rate of 1 ml min$^{-1}$. Samples were monitored by an ultraviolet detector at 245 nm wavelength. For monoamines quantification, the mobile phase is a mixed solution consisting of OSA (1.7 mM), NaH$_2$PO$_4$·2H$_2$O (90.0 mM), C$_6$H$_8$O$_7$·H$_2$O (50.0 mM), EDTA·2Na (50.0 μM) and 5% acetonitrile with the flow rate of 0.2 ml min$^{-1}$. Samples were monitored by an electrochemical detector at 350 mV. For amino acids quantification, the mobile phase is Na$_2$HPO$_4$·12H$_2$O (40 mM) and methyl alcohol with overall flow rate of 1 ml min$^{-1}$. Samples were monitored by fluorescence detector at 328 nm excitation wavelength and 425 nm emission wavelength.

**Behavioral assays**. Behavioral assays were performed 3 days after the final injection of MPTP or saline. For locomotor activity test, mice were placed in an activity monitor chamber (20 cm × 20 cm × 15 cm) to acclimatize for 15 min before the start of the test. The track of the mice was recorded in 5 min and the speed was calculated using Open field software (Clever Sys Inc., VA, USA). For rotarod test, mice were placed in a separate compartment on the rod and tested at 6 × g for 5 min before acclimatized to the rod with a fixed speed of 3.5 × g for 5 min for 2 consecutive days. The latency to fall was recorded using Rotarod Analysis System (Jiliang, Shanghai, China). For the pole test, mice were placed on the top of the wooden pole (diameter 1 cm, height 50 cm, rough surface). The time taken by the mice to reach the floor with its four paws was recorded as locomotion activity time T-total and the time taken by the mice to turn completely head downward was recorded T-turn time. Each mouse was accustomed to the apparatus on the day before testing. The test was performed three times to ensure accuracy.

**Statistics**. Statistical significance was determined using GraphPad Prism 7; Student's unpaired two-tailed t-test, one-way ANOVA or two-way ANOVA was conducted according to test requirements. *$P < 0.05$, **$P < 0.01$, and ***$P < 0.001$ were considered significant. The number of replicates and repeats of individual experiments and statistical tests is indicated in the legends.

**Reporting summary**. Further information on research design is available in the Nature Research Reporting Summary linked to this article.

## Data availability
The RNA sequenceing data that support the findings of this study have been deposited at the NCBI GEO data repository under accession number GSE143164. Source data underlying Figs. 1–6 and Supplementary Figs. 1–8 is available as a Source Data file. A detailed description of all statistical analyses performed for Figs. 1–6 and Supplementary

Figs. 1–8, including a brief description for each figure, sample size, statistical tests performed, P values, post hoc test, and post hoc test P values were provided as a Source Data file. Other data that support the findings of this study are available from the corresponding author upon reasonable request.

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

## Acknowledgements

The authors thank Peng Cao from Affiliated Hospital of Integrated Traditional Chinese and Western Medicine, Nanjing University of Chinese Medicine, for providing *Nrf2*-knockout mice; Jiawei Zhou from Shanghai Institutes for Biological Science, Chinese Academy of Sciences, for providing *hGFAP-Cre* mice, *Drd2flox/flox* mice, and *Drd2*-knockout mice; Gang Pei from Shanghai Key laboratory of Signaling and Disease Research, Tongji University, for providing *β-arrestin2*-knockout mice. The following grants support this research: the National Natural Science Foundation of China (No. 81630099, No. 81991523 and No. 81703491), the Drug Innovation Major Project (No. 2018ZX09711001-003-007), and the Fok Ying Tung Education Foundation (No. 20173237210003).

## Author contributions

G.H., Y.W., and M.L. designed the experiments. Y.W. and M.M. performed most of the in vivo and in vitro experiments with the help of H.R.W. Z.T.H. and H.Y. performed the UHPLC assay. M.M.C. and N.S.S. constructed ADV vectors. X.D. and J.H.D. contributed to genotyping. M.X. performed proteomics processing and analysis. G.H. and Y.W. analyzed data and wrote the manuscript. G.H. is the leading principal investigator who supervised this research and edited the paper.

## Competing interests

The authors declare no competing interests.
