## [Peer Review File · Nature Communications]

Reviewers' comments:

Reviewer #1 (Remarks to the Author):

Wei and colleagues in their manuscript entitled "Astrocytic DRD2 induced glutathione synthesis via PKM2-mediated Nrf2 transactivation", report a novel mechanism by which anti-parkinsonian DA agonists induce brain glutathione biosynthesis. They show that DA agonists act via astrocytic D2 receptor (DRD2) to induce GSH synthesis via a mechanism involving M2 pyruvate kinase (PKM2)-mediated Nrf2 transactivation. Following DRD2 activation, through a G-protein coupled receptor mechanism, beta-arrestin 2 binds to the tetrameric PKM2 resulting in the formation of PKM2 dimer which then interacts with the antioxidant transcription factor Nrf2 to function as a transcriptional coactivator, resulting in the binding of Nrf2 to the promoter of glutamate-cysteine ligase and subsequently increase GSH synthesis. They further performed a natural product screen targeting PKM2 and found Vit B6 (pyridoxine) which can dimerize PKM2 to promote GSH biosynthesis. Furthermore, pyridoxine treatment in mice protected against MPTP-induced nigral dopaminergic toxicity both in wild type and astrocytic DRD2-conditional knockout mice. Overall, the results demonstrated that pyridoxine can induce GSH synthesis to protect nigral DA neurons in a DRD2 receptor-independent manner. This is a well-conceived manuscript demonstrating a novel mechanism involving the dimerization of PKM2 to activate the protective Nrf2 signaling pathway and should be of significant interest to the community. Many of the experiments are carefully done and technically sound. The methods and experimental details are well described in the method section and figure legends and there is appropriate use of statistical methods. The major weakness of this study is the lack of data demonstrating terminal protection in the MPTP model at the level of striatal DA and its metabolites with pyridoxine and other DA agonists in the MPTP model. Another major weakness is the method used to perform stereological cell counts of the TH neurons in the MPTP studies where Nissl- and TH-positive neuronal counts were not performed on the same sections questioning the validity of the morphometric measurements. Also, striatal MPP⁺ levels were not measured to rule out the possibility that the protective effects of pyridoxine against MPTP toxicity is not due to the lack of conversion of MPTP to MPP⁺ which takes place in the astrocytes. Lastly, pyridoxine is known to cause neuropathy in humans, a significant side-effect, and the authors didn't address this issue in this manuscript.

Reviewer #2 (Remarks to the Author):

A variety of dopamine agonists are well known to increase brain glutathione levels, and the authors set out to define the mechanisms involved. They identify Nrf2 activation as the mechanism, induced by a non-canonical signaling pathway involving PKM2 dimer formation and association with Nrf2 as a coactivator. They identify pyridoxine as a small molecule that can cause PKM2 dimerisation, activate Nrf2, induce GSH synthesis and protect in the MPTP PD model. Although this is a potentially interesting pathway of activation for Nrf2, I am not convinced that it represents a sufficient conceptual advance for readers of this journal.

The authors do not cite a key paper where dopamine agonists have been previously reported to activate the Nrf2 pathway in astrocytes, boosting metabolic flux through the pentose-phosphate pathway, the primary mechanism for generating NADPH to recycle glutathione to the reduced form (PMID: 29768946).

Astrocytic Nrf2 is a well established target in preclinical PD research. Transgenic mice over-expressing Nrf2 specifically in astrocytes are protected from neurotoxicity in the MPTP model of PD, and delays pathology and motor deficits in the alpha synuclein A53T mutant mouse model (PMID: 19196989, 23223297). Several small molecule activators of the Nrf2 pathway offer protection in the MPTP model of PD, with astrocytes implicated (e.g. PMID 30739428, 28986252, 28167258, 24991814).

While the authors report a new pathway of activation for Nrf2, non-canonical mechanisms of Nrf2 activation in astrocytes have been described, e.g. MAPK, P75-NTR, AMPK, TNF α (PMID: 30845718, 30739428, 29374533, 22941452). Moreover, the strength of activation seems very modest.

The compound that they identify pyridoxine, has been previously identified as an activator of the Nrf2 pathway (PMID 30084803, 28912874, 30698872). There also seems to be a lack of discussion that pyridoxine (Vitamin B6) is consumed by the metabolism of L-DOPA, and is often supplemented to PD patients. Importantly, circulating pyridoxine must be kept tightly regulated, as too much peripherally results in peripheral metabolism of L-DOPA and lack of bioavailability in the CNS. Thus, high dose of pyridoxine are not an option therapeutically.

Specific Points:

Figure 1a. the increase in GSH levels is very modest-what is the 95% CI of the increase? Is this likely to be biologically significant?

Figure 1 (and throughout) There is no information in the legends regarding the statistical test employed so it is impossible to tell if it was appropriate or not. Even the methods section, which mentions Anovas, makes no mention of any post-hoc tests.

Figure 1 (and throughout) employ incorrect statistical comparisons, since they make the basic mistake of rejecting the null hypothesis based on two tests being either side of a p-value threshold. Such erroneous analyses of interactions have been highlighted in nature journals previously (see PMID 21878926).

Figure 1i. How does this level of induction of Nrf2 targets compare to classical Nrf2 activators? Molecules like tBHQ induce Hmox1 in astrocytes by tens to hundreds of fold, whereas induction here is very modest (and in the case of Hmox1, insignificant)

Fig. 2a. I find it hard to see how this Nrf2 antibody can be so successful in picking up proteins by co-IP when it can barely detect Nrf2 in the western blot (Fig. 1j) Can some conventional co_IP experiments be shown early on to confirm the interaction?

Fig. 2c No quantitation is shown

Fig. 3. It is unclear what the control is for the Pkm2 siRNA experiments, since there seems to be no mention of any control siRNA in the legends or methods.

Fig. 4. Pyridoxine can increase brain dopamine levels (PMID 30413185) and is the cofactor for l-aromatic amino acid decarboxylase, which converts brain L-DOPA to dopamine. Could it be protecting by increasing dopamine levels and thus competing out MPP+ uptake via the dopamine transporter or other mechanisms? An astrocyte-specific Nrf2 KO would be needed to show the mechanism involved is in line with the authors' model, since pyridoxine drives many pathways.

Fig. 4. There seems to be a lack of discussion that pyridoxine (Vitamin B6) is consumed by the metabolism of L-DOPA, and is often supplemented to PD patients. Importantly, circulating pyridoxine must be kept tightly regulated, as too much peripherally results in peripheral metabolism of L-DOPA and lack of bioavailability in the CNS. Thus, high dose of pyridoxine are not an option therapeutically.

Reviewer #3 (Remarks to the Author):

Wei and collaborators present a beautiful paper on the mechanism whereby astrocytic DRD2 induces GSH biosynthesis through PKM2 dimerization, which then binds to and activates Nrf2, which is an antioxidant mechanism. Moreover, the authors show that pyridoxine facilitates GSH synthesis through the PKM2-Nrf2 pathway in D2DR-independent fashion. The authors end up by proposing this drug as a strategy to break the vicious cycle of "less dopamine release – GSH synthesis deficiency – oxidative measure – dopaminergic neuron death" in the progression of PD. However, the authors need to clarify the following items prior to the publication of this manuscript:

1-What is the contribution (if any) of PKM2 in microglia to the pyridoxine effect in vivo?

2-We know that pyridoxine is a co-factor of DOPA-carboxylase, thus enhancing the systemic conversion of dopa to dopamine. Moreover, this vitamin is also a cofactor for both glutamic acid decarboxylase and GABA transaminase, which are required for the synthesis and metabolism of GABA in the brain. Therefore, how can you unequivocally ascertain the role of direct dimerization of PKM2 as the sole contributor to the in vivo effect of pyridoxine?

3-How did you reach dose and duration of treatment? The authors treated the animals continuously with 5 mg/kg quinpirole or 5 mg/kg pyridoxine for 7 (or 8?) consecutive days. How did they reach these dosing protocols? It would be helpful to display a schematic representation of the in vivo protocol including dosing and time of sacrifice.

4-I am curious to know which was the impact of both quinpirole and pyridoxine on MPTP-induced striatal astrogliosis and locomotor impairment.

Point-to-Point Response to Reviewers

Reviewer #1

Wei and colleagues in their manuscript entitled "Astrocytic DRD2 induced glutathione synthesis via PKM2-mediated Nrf2 transactivation", report a novel mechanism by which anti-parkinsonian DA agonists induce brain glutathione biosynthesis. They show that DA agonists act via astrocytic D2 receptor (DRD2) to induce GSH synthesis via a mechanism involving M2 pyruvate kinase (PKM2)-mediated Nrf2 transactivation. Following DRD2 activation, through a G-protein coupled receptor mechanism, beta-arrestin 2 binds to the tetrameric PKM2 resulting in the formation of PKM2 dimer which then interacts with the antioxidant transcription factor Nrf2 to function as a transcriptional coactivator, resulting in the binding of Nrf2 to the promoter of glutamate-cysteine ligase and subsequently increase GSH synthesis. They further performed a natural product screen targeting PKM2 and found Vit B6 (pyridoxine) which can dimerize PKM2 to promote GSH biosynthesis. Furthermore, pyridoxine treatment in mice protected against MPTP-induced nigral dopaminergic toxicity both in wild type and astrocytic DRD2-conditional knockout mice. Overall, the results demonstrated that pyridoxine can induce GSH synthesis to protect nigral DA neurons in a DRD2 receptor-independent manner. This is a well-conceived manuscript demonstrating a novel mechanism involving the dimerization of PKM2 to activate the protective Nrf2 signaling pathway and should be of significant interest to the community. Many of the experiments are carefully done and technically sound. The methods and experimental details are well described in the method section and figure legends and there is appropriate use of statistical methods.

Major comments:

1. The major weakness of this study is the lack of data demonstrating terminal protection in the MPTP model at the level of striatal DA and its metabolites with pyridoxine and other DA agonists in the MPTP model.

Answer: We greatly appreciate the reviewer's comment. We have performed ultra high-performance liquid chromatography (UHPLC) to detect the level of striatal DA, DOPAC and GABA with quinpirole or pyridoxine treatment in the MPTP model. Both quinpirole and pyridoxine administration prevented MPP⁺-induced loss of nigral DA (Supplementary Fig. 8I) in *Drd2^{fllox/fllox}* mice. In contrast, *Drd2^{hGFAP}cKO* mice showed only attenuation of nigral DA loss induced by MPP⁺ following pyridoxine treatment but not alleviated DA loss following quinpirole treatment (Supplementary Fig. 8I). These results indicate that pyridoxine provides terminal protection when administered before MPTP independent of dopamine receptors.

2. Another major weakness is the method used to perform stereological cell counts of the TH neurons in the MPTP studies where Nissl- and TH-positive neuronal counts were not performed on the same sections questioning the validity of the morphometric measurements.

Answer: Thanks for the reviewer's comment. Nissl- and TH-positive neuronal counts were performed on 12 sections in the midbrain of each mouse to exclude the possible error produced by counting only one section of the midbrain. We have shown the typical SNc sections in Figure 6g and Supplementary Figure 8f.

3. Also, striatal MPP⁺ levels were not measured to rule out the possibility that the protective

effects of pyridoxine against MPTP toxicity is not due to the lack of conversion of MPTP to MPP⁺ which takes place in the astrocytes.

Answer: Thanks for the reviewer's comment. As shown in Supplementary Fig. 8e, pyridoxine treatment did not alter the toxic MPP⁺ levels in the striatum, indicating that the protective effects of pyridoxine against MPTP toxicity were not due to the lack of conversion of MPTP to MPP⁺.

4. Lastly, pyridoxine is known to cause neuropathy in humans, a significant side-effect, and the authors didn't address this issue in this manuscript

Answer: Thanks for the reviewer's comment. We discussed this issue in line 301-307. The current Recommended Dietary Allowance (RDA) from the American National Institute of Health (NIH) for pyridoxine is 2 mg per day with an upward tolerance of 100 mg per day for adults (PMID: 19145213). 5 mg/kg pyridoxine treatment can induce GSH synthesis efficiently in mice. According to the body surface area normalization method described by FDA draft guidelines (PMID: 30343496), the human equivalent dose is about 32 mg per day for an 80kg adult. Thus, we postulate that pyridoxine is a relatively safe candidate for a therapeutical option.

Reviewer #2

A variety of dopamine agonists are well known to increase brain glutathione levels, and the authors set out to define the mechanisms involved. They identify Nrf2 activation as the mechanism, induced by a non-canonical signaling pathway involving PKM2 dimer formation and association with Nrf2 as a coactivator. They identify pyridoxine as a small molecule that can cause PKM2 dimerisation, activate Nrf2, induce GSH synthesis and protect in the MPTP PD model. Although this is a potentially interesting pathway of activation for Nrf2, I am not convinced that it represents a sufficient conceptual advance for readers of this journal.

We feel sorry that we did not express well about our novelty and significance in our previous manuscript, which caused a misunderstanding about our conceptual advance for the reviewer. The key point of our work is not that we found a potential way to activate Nrf2.

Oxidative stress, defined as the overproduction of ROS, is widely considered a major pathogenic mechanism of PD. To sustain ROS homeostasis, our brains employ GSH as an important antioxidant to protect neurons from oxidative damage. Unfortunately, accumulating evidence indicates that GSH is greatly decreased in the brains of PD patients. Hence, clarifying the mechanism underlying the decline in GSH and trying to restore the GSH level are very important in PD treatment.

In our study, we demonstrate that the astrocytic dopamine D2 receptor (DRD2) induces GSH synthesis via PKM2-mediated Nrf2 transactivation, which indicates the loss of dopaminergic neurons would start a vicious cycle of “decreased dopamine release—DRD2 signaling deficiency—decreased GSH synthesis—oxidative stress—dopaminergic neuron death” in the progression of PD and explain the mechanism by which GSH decreases in the brains of PD patients. We think clarifying the mechanism underlying the decline of GSH in PD is the first point of our work.

According to our results, we can activate astrocytic DRD2, PKM2, or Nrf2 to restore GSH levels in PD. DRD2 and Nrf2 are expressed in both neurons and astrocytes. However, PKM2

is predominantly expressed in astrocytes. The low expression of PKM2 in neurons provides an advantage because chemicals targeting PKM2 can act exclusively on astrocytes, preventing the side effects induced by affecting neuron functions. Hence, we selected PKM2 as a target for GSH recovery in PD and further screened pyridoxine as a chemical to activate the PKM2-Nrf2 pathway. We think identifying PKM2 as a potential target in PD treatment is the second point of our work.

Major comments:

1. The authors do not cite a key paper where dopamine agonists have been previously reported to activate the Nrf2 pathway in astrocytes, boosting metabolic flux through the pentose-phosphate pathway, the primary mechanism for generating NADPH to recycle glutathione to the reduced form (PMID: 29768946)

Answer: Thanks for the reviewer's comment, which is a crucial comment concerning the novelty of our manuscript. We want to reply to this concern as follows:

First, the paper (PMID: 29768946) didn't directly demonstrate that dopamine activates the Nrf2 pathway. This paper reported dopamine exposure to astroglia enhanced the transcription of HO-1, whose transcription is regulated by Nrf2. But the author only detected the mRNA level of HO-1 by qPCR. Other mechanisms independent of Nrf2 may lead to the same phenomenon. For example, if dopamine inhibits HO-1 mRNA degradation, the HO-1 mRNA levels will also increase. Without a ChIP assay to detect Nrf2 binding to the promoter of HO-1, this paper can not draw a conclusion that dopamine activates the Nrf2 pathway.

Second, we know Nrf2 is a transcription factor of G6PDH, which is the rate-limiting enzyme of the pentose-phosphate pathway (PPP). However, dopamine-induced PPP activation cannot demonstrate that dopamine activates the Nrf2 pathway because dopamine may affect PPP through other pathways. For example, if dopamine does not increase the level of G6PDH but enhances the enzyme activity of G6PDH, we can also detect a boosting metabolic flux into PPP. Since the title of this paper is "Neuroprotective Role of Astroglia in Parkinson Disease by Reducing Oxidative Stress Through Dopamine-Induced Activation of Pentose-Phosphate Pathway", the author did not conclude that they demonstrate dopamine can activate the Nrf2 pathway.

Third, even we agree that dopamine activates Nrf2 has been reported in this paper, the molecule mechanism remains unknown. Our results that dopamine induces PKM2 dimerization to activate Nrf2 can provide explanations for this phenomenon.

2. Astrocytic Nrf2 is a well established target in preclinical PD research. Transgenic mice over-expressing Nrf2 specifically in astrocytes are protected from neurotoxicity in the MPTP model of PD, and delays pathology and motor deficits in the alpha synuclein A53T mutant mouse model (PMID: 19196989, 23223297). Several small molecule activators of the Nrf2 pathway offer protection in the MPTP model of PD, with astrocytes implicated (e.g. PMID 30739428, 28986252, 28167258, 24991814). While the authors report a new pathway of activation for Nrf2, non-canonical mechanisms of Nrf2 activation in astrocytes have been described, e.g. MAPK, P75-NTR, AMPK, TNF α (PMID: 30845718, 30739428, 29374533, 22941452). Moreover, the strength of activation seems very modest.

Answer: Thanks for the reviewer's comment. The novelty of our work is not that we found a new way to activate Nrf2. The key point of our work is that we identified PKM2 as a potential

target in PD treatment. We have discussed why PKM2 may be a better target than Nrf2 for GSH recovery in PD in line 188-203. Since Nrf2 expresses in both neurons and astrocytes, chemicals targeting Nrf2 may affect the physiological functions of neurons and result in unexpected side effects. However, PKM2 is predominantly expressed in astrocytes. The low expression of PKM2 in neurons provides an advantage because chemicals targeting PKM2 can act exclusively on astrocytes, preventing the side effects induced by affecting neuron functions. Hence, we selected PKM2 as a target for GSH recovery in PD.

3. The compound that they identify pyridoxine, has been previously identified as an activator of the Nrf2 pathway (PMID 30084803, 28912874, 30698872). There also seems to be a lack of discussion that pyridoxine (Vitamin B6) is consumed by the metabolism of L-DOPA, and is often supplemented to PD patients. Importantly, circulating pyridoxine must be kept tightly regulated, as too much peripherally results in peripheral metabolism of L-DOPA and lack of bioavailability in the CNS. Thus, high dose of pyridoxine are not an option therapeutically.

Answer: Thanks for the reviewer's comment. Those three papers reported that pyridoxine is an activator of the Nrf2 pathway, which is in accordance with our results that pyridoxine activates Nrf2 to induce GSH synthesis. The novelty of our work is that we demonstrate that pyridoxine can dimerize PKM2 to coactivate Nrf2, which explains how pyridoxine activates Nrf2.

We also discussed the relationship between pyridoxine and L-DOPA in line 291-299. Madopar, which is a mixture of L-DOPA and benserazide, is the most widely used drug in PD treatment. As an inhibitor of dopa decarboxylase, benserazide, which can not penetrate the blood-brain-barrier, helps to inhibit the pyridoxine induced-metabolism of L-DOPA in peripheral and enhance the bioavailability of L-DOPA in the CNS. Recent studies have demonstrated that a pyridoxine intake of at most 50 mg/day would not reduce the bioavailability of L-DOPA in the CNS, and a proper supplementation of pyridoxine can prevent L-DOPA-induced symptomatic pyridoxine deficiency. These researches all support pyridoxine as a therapeutical option.

Specific Points:

1. Figure 1a. the increase in GSH levels is very modest-what is the 95% CI of the increase? Is this likely to be biologically significant?

Answer: Thanks for the reviewer's comment. The 95% CI of the increase is shown below. We performed 6 independent experiments, and the increase is likely to be biologically significant.

Dunnett's multiple comparisons test	Mean Diff.	95.00% CI of diff.	Significant?	Summary	Adjusted P Value
Astrocyte					
0 vs. 5	-0.2160	-2.231 to 1.799	No	ns	0.9985
0 vs. 10	-3.886	-5.901 to -1.871	Yes	****	<0.0001
0 vs. 20	-3.355	-5.370 to -1.340	Yes	***	0.0003
0 vs. 40	-6.183	-8.198 to -4.168	Yes	****	<0.0001
0 vs. 80	-3.446	-5.461 to -1.431	Yes	***	0.0002

2. Figure 1 (and throughout) There is no information in the legends regarding the statistical test employed so it is impossible to tell if it was appropriate or not. Even the methods section, which mentions Anovas, makes no mention of any post-hoc tests.

Answer: Thanks for the reviewer's comment. We add the information about statistical tests and post-hoc tests at the end of each legend.

3. Figure 1 (and throughout) employ incorrect statistical comparisons, since they make the basic mistake of rejecting the null hypothesis based on two tests being either side of a p-value threshold. Such erroneous analyses of interactions have been highlighted in nature journals previously (see PMID 21878926).

Answer: Thanks for the reviewer's comment. We have read the paper carefully and employed the proper statistical comparisons in each figure.

4. Figure 1i. How does this level of induction of Nrf2 targets compare to classical Nrf2 activators? Molecules like tBHQ induce Hmox1 in astrocytes by tens to hundreds of fold, whereas induction here is very modest (and in the case of Hmox1, insignificant)

Answer: Thanks for the reviewer's comment. First, Figure 1i was designed to validate whether DRD2 activation induces the Nrf2 pathway. We are not aimed to find a stronger Nrf2 activator here. Second, as shown below, Fang Ye et.al. treated SH-SY5Y cells with 40 uM tBHQ and saw a similar modest increase of *Gclc* to our results (PMID: 26798413). Third, quinpirole acts on DRD2 and activates Nrf2 in an indirect way. Hence it is reasonable that quinpirole induces less increase of Nrf2 targets than tBHQ, which activates Nrf2 directly.

[REDACTED]

5. Fig. 2a. I find it hard to see how this Nrf2 antibody can be so successful in picking up proteins by co-IP when it can barely detect Nrf2 in the western blot (Fig. 1j) Can some conventional co_IP experiments be shown early on to confirm the interaction?

Answer: Thanks for the reviewer's comment. Our Nrf2 antibody (Proteintech, 66504-1-Ig) has been widely used and validated in other publications (PMID: 30950217, PMID: 31212129, PMID: 31285477). The shallow WB bands of Nrf2 is due to that Nrf2 is not a high expression protein in cells. Hence, we prepared about 600 ug protein for each co-IP experiment to make

sure that we get enough IP products for further analysis. Besides, we also employed the proximity ligation assay to validate our co-IP results.

6. Fig. 2c No quantitation is shown

Answer: Thanks for the reviewer's comment. We quantitate the PLA signal in 50 cells per group and show the results below.

7. Fig. 3. It is unclear what the control is for the Pkm2 siRNA experiments, since there seems to be no mention of any control siRNA in the legends or methods.

Answer: Thanks for the reviewer's comment. We used negative control siRNA in the PKM2 siRNA experiments. We have added the information to the methods in line 372.

8. Fig. 4. Pyridoxine can increase brain dopamine levels (PMID 30413185) and is the cofactor for l-aromatic amino acid decarboxylase, which converts brain L-DOPA to dopamine. Could it be protecting by increasing dopamine levels and thus competing out MPP+ uptake via the dopamine transporter or other mechanisms? An astrocyte-specific Nrf2 KO would be needed to show the mechanism involved is in line with the authors' model, since pyridoxine drives many pathways.

Answer: Thanks for the reviewer's comment. We know that pyridoxine is the coenzyme of dopa decarboxylase and GABA transaminase. Altered metabolism of dopamine and GABA may contribute to *in vivo* effects of pyridoxine. Thus, we performed ultra high-performance liquid chromatography (UHPLC) to detect the level of striatal DA and GABA with pyridoxine treatment. Surprisingly, as shown in Supplementary Fig. 8l, n, we found pyridoxine treatment did not alter the striatal DA and GABA, which is in accordance with a previous publication (PMID 30413185, shown below). These results are probably due to that the enzyme activity of dopa decarboxylase and GABA transaminase are relative redundant and are not the rate-limiting step for the metabolism of DA and GABA.

Though pyridoxine does not alter the levels of DA and GABA in the striatum, we still can not

exclude that pyridoxine may also confer neuroprotection through other undiscovered mechanisms. Thus, we state this limitation in line 207-271.

For we do not have astrocyte-specific Nrf2 KO mice, we treated *Nrf2*-knockout mice with pyridoxine. As shown in Supplementary Fig. 8b, ablation of *Nrf2* attenuated pyridoxine induced GSH synthesis, indicating that pyridoxine facilitates GSH synthesis at least partially through Nrf2 *in vivo*.

[REDACTED]

9. Fig. 4. There seems to be a lack of discussion that pyridoxine (Vitamin B6) is consumed by the metabolism of L-DOPA, and is often supplemented to PD patients. Importantly, circulating pyridoxine must be kept tightly regulated, as too much peripherally results in peripheral metabolism of L-DOPA and lack of bioavailability in the CNS. Thus, high dose of pyridoxine are not an option therapeutically.

Answer: Thanks for the reviewer's comment. We have discussed the relationship between pyridoxine and L-DOPA in line 291-299. Madopar, which is a mixture of L-DOPA and benserazide, is the most widely used drug in PD treatment. As an inhibitor of dopa decarboxylase, benserazide, which can not penetrate the blood-brain-barrier, helps to inhibit the pyridoxine induced-metabolism of L-DOPA in peripheral and enhance the bioavailability of L-DOPA in the CNS. Recent studies have demonstrated that a pyridoxine intake of at most 50 mg/day would not reduce the bioavailability of L-DOPA in the CNS, and a proper supplementation of pyridoxine can prevent L-DOPA-induced symptomatic pyridoxine deficiency. These researches all support pyridoxine as a therapeutical option.

Reviewer #3:

Wei and collaborators present a beautiful paper on the mechanism whereby astrocytic DRD2 induces GSH biosynthesis through PMK2 dimerization, which then binds to and activates Nrf2, which is an antioxidant mechanism. Moreover, the authors show that pyridoxine facilitates GSH synthesis through the PKM2-Nrf2 pathway in D2DR-independent fashion. The authors end up by proposing this drug as a strategy to break the vicious cycle of “less

dopamine release – GSH synthesis deficiency – oxidative measure – dopaminergic neuron death” in the progression of PD. However, the authors need to clarify the following items prior to the publication of this manuscript:

Major comments:

1. What is the contribution (if any) of PKM2 in microglia to the pyridoxine effect *in vivo*?

Answer: Thanks for the reviewer’s comment. As shown below, microglia express mainly PKM2 (a) and pyridoxine induces GSH synthesis in microglia (b), indicating pyridoxine may also induce GSH synthesis through microglia *in vivo*. However because astrocytes synthesize and release much more GSH than microglia (PMID: 12125073) and astrocytes form the highest population of glial cells in the CNS (PMID: 1603325), we speculate that pyridoxine may induce GSH synthesis mainly through astrocytes. Besides, astrocytes interact closely with neurons and enwrap neuronal components to form tripartite synapse, which provides an efficient way to transfer GSH into neurons (PMID: 22417747). For the above reasons, we think PKM2 in astrocyte makes major contributions to the pyridoxine effect *in vivo*.

2. We know that pyridoxine is a co-factor of DOPA-carboxylase, thus enhancing the systemic conversion of dopa to dopamine. Moreover, this vitamin is also a cofactor for both glutamic acid decarboxylase and GABA transaminase, which are required for the synthesis and metabolism of GABA in the brain. Therefore, how can you unequivocally ascertain the role of direct dimerization of PKM2 as the sole contributor to the *in vivo* effect of pyridoxine?

Answer: Thanks for the reviewer’s comment. We know that pyridoxine is the coenzyme of dopa decarboxylase and GABA transaminase. Altered metabolism of dopamine and GABA may contribute to *in vivo* effects of pyridoxine. Thus, we performed ultra high-performance liquid chromatography (UHPLC) to detect the level of striatal DA and GABA with pyridoxine treatment. Surprisingly, as shown in Supplementary Fig. 8l, n, we found pyridoxine treatment did not alter the striatal DA and GABA, which is in accordance with a previous publication (PMID 30413185). These results are probably due to that the enzyme activity of dopa decarboxylase and GABA transaminase are relative redundant and are not the rate-limiting step for the metabolism of DA and GABA.

Though pyridoxine does not alter the levels of DA and GABA in the striatum, we still can not exclude that pyridoxine may also confer neuroprotection through other undiscovered mechanisms. Thus, we state this limitation in line 207-271. We think the neuroprotective

effect of pyridoxine is at least partially due to its ability to dimerize PKM2 and activate Nrf2.

3. How did you reach dose and duration of treatment? The authors treated the animals continuously with 5 mg/kg quinpirole or 5 mg/kg pyridoxine for 7 (or 8?) consecutive days. How did they reach these dosing protocols? It would be helpful to display a schematic representation of the *in vivo* protocol including dosing and time of sacrifice.

Answer: Thanks for the reviewer's comment. The dose and duration of quinpirole treatment were chosen according to our previous publications (PMID: 23242137, PMID: 29786071). We also treated wild-type mice with different concentrations of pyridoxine for 7 consecutive days to determine the effective pyridoxine concentration for inducing GSH synthesis *in vivo*. As shown in Supplementary Fig. 8a, 5 mg/kg pyridoxine increased GSH levels as efficiently as quinpirole.

Thanks for the reviewer's comment. We have added a schematic representation of the *in vivo* protocol including dosing and time of sacrifice in Supplementary Fig. 8c.

4. I am curious to know which was the impact of both quinpirole and pyridoxine on MPTP-induced striatal astrogliosis and locomotor impairment.

Answer: Answer: Thanks for the reviewer's comment. As shown below, both quinpirole and pyridoxine administration prevented MPTP-induced astrogliosis in *Drd2^{lox/lox}* mice. In contrast, *Drd2^{hGFAP}cKO* mice showed only attenuation of astrogliosis induced by MPTP following pyridoxine treatment but not alleviated astrogliosis following quinpirole treatment. We also evaluated locomotor impairment in *Drd2^{lox/lox}* mice and *Drd2^{hGFAP}cKO* mice after quinpirole or pyridoxine treatment in the MPTP mouse model. Pyridoxine treatment shortened the time for turning around (T-Turn) and time for descending a pole (T-TLA) of both *Drd2^{lox/lox}* mice and *Drd2^{hGFAP}cKO* mice in the pole test (Supplementary Fig. 8h, i) but did not alter the latency time in the rotarod test (Supplementary Fig. 8j) or the activity of mice in the open field test (Supplementary Fig. 8k), indicating that pyridoxine can improve some types of mouse locomotion in a DRD2-independent manner.

REVIEWERS' COMMENTS:

Reviewer #1 (Remarks to the Author):

The authors were very responsive to the prior critiques and the quality of the manuscript has now significantly improved.

Reviewer #2 (Remarks to the Author):

The authors do not dispute that pyridoxine is known to activate Nrf2, nor that DA agonists can induce Nrf2 and activate GSH production in astrocytes. However they show that Nrf2 activation by DA agonists is via the novel mechanism of PKM2 dimerization, and make it clearer that this is the primary novel aspect of the paper. Moreover, they make the case that since PKM2 is highly enriched in astrocytes relative to neurons, drugs targeting PKM2 may preferentially target astrocytes. This is not proven though, and would require (for example) conditional deletion or knockdown of PKM2 in astrocytes. Also it is unclear why direct Nrf2 activators may not be a better strategy, since all glial types express Nrf2, and it is only very weakly active in neurons. Overall they strengthen their case around the novelty of the paper, though direct demonstration of the importance of PKM2 in astrocytes for the protective effect of pyridoxine in vivo would nail it.

Reviewer #3 (Remarks to the Author):

The authors fully answered to my questions. I believe that the authors clearly made an effort to improve the quality of the ms.

Frederico C. Pereira

Point-to-Point Response to Reviewers

Reviewer #1

The authors were very responsive to the prior critiques and the quality of the manuscript has now significantly improved.

Answer: We greatly appreciate the reviewer's comment for helping us improve our manuscript.

Reviewer #2

The authors do not dispute that pyridoxine is known to activate Nrf2, nor that DA agonists can induce Nrf2 and activate GSH production in astrocytes. However they show that Nrf2 activation by DA agonists is via the novel mechanism of PKM2 dimerization, and make it clearer that this is the primary novel aspect of the paper. Moreover, they make the case that since PKM2 is highly enriched in astrocytes relative to neurons, drugs targeting PKM2 may preferentially target astrocytes. This is not proven though, and would require (for example) conditional deletion or knockdown of PKM2 in astrocytes. Also it is unclear why direct Nrf2 activators may not be a better strategy, since all glial types express Nrf2, and it is only very weakly active in neurons. Overall they strengthen their case around the novelty of the paper, though direct demonstration of the importance of PKM2 in astrocytes for the protective effect of pyridoxine *in vivo* would nail it.

Answer: We greatly appreciate the reviewer's comment for helping us improve our manuscript.

The well-known Nrf2 activator tBHQ has been reported to be a double-edged sword (PMID: 22466069). tBHQ can exert cytoprotection caused by Nrf2-mediated induction of genes involved in antioxidative defense. However, some researchers reported that tBHQ leads to tumor formation in rodents for unknown mechanisms. Besides, tBHQ was also reported to have cytotoxicity because it perturbs the redox cycling processes. Hence, we regard pyridoxine targeting PKM2 as a relatively safer and better candidate for PD treatment.

According to our results, the expression of PKM2 is low in the neurons of the mouse brain, which coincides with its expression in primary neurons. Since pyridoxine failed to induce Nrf2 activation in primary neurons (Fig. 5m and n), we postulate that pyridoxine may preferentially target PKM2 in astrocytes rather than those in neurons *in vivo*. According to those experiments, we tried our best to demonstrate that drugs targeting PKM2 may preferentially target astrocytes *in vivo* with the limitation of lacking astrocytic PKM2 conditional knockout mice.

Reviewer #3

The authors fully answered to my questions. I believe that the authors clearly made an effort to improve the quality of the ms.

Answer: We greatly appreciate the reviewer's comment for helping us improve our manuscript.